# Biophysically grounded mean-field models of neural populations under electrical stimulation

**Caglar Cakan**[1,2]*, **Klaus Obermayer**[1,2]

**1** Department of Software Engineering and Theoretical Computer Science, Technische Universität Berlin, Germany, **2** Bernstein Center for Computational Neuroscience Berlin, Germany

* cakan@ni.tu-berlin.de

## Abstract

Electrical stimulation of neural systems is a key tool for understanding neural dynamics and ultimately for developing clinical treatments. Many applications of electrical stimulation affect large populations of neurons. However, computational models of large networks of spiking neurons are inherently hard to simulate and analyze. We evaluate a reduced mean-field model of excitatory and inhibitory adaptive exponential integrate-and-fire (AdEx) neurons which can be used to efficiently study the effects of electrical stimulation on large neural populations. The rich dynamical properties of this basic cortical model are described in detail and validated using large network simulations. Bifurcation diagrams reflecting the network's state reveal asynchronous up- and down-states, bistable regimes, and oscillatory regions corresponding to fast excitation-inhibition and slow excitation-adaptation feedback loops. The biophysical parameters of the AdEx neuron can be coupled to an electric field with realistic field strengths which then can be propagated up to the population description. We show how on the edge of bifurcation, direct electrical inputs cause network state transitions, such as turning on and off oscillations of the population rate. Oscillatory input can frequency-entrain and phase-lock endogenous oscillations. Relatively weak electric field strengths on the order of 1 V/m are able to produce these effects, indicating that field effects are strongly amplified in the network. The effects of time-varying external stimulation are well-predicted by the mean-field model, further underpinning the utility of low-dimensional neural mass models.

## Author summary

Weak electrical inputs to the brain *in vivo* using transcranial electrical stimulation or in isolated cortex *in vitro* can affect the dynamics of the underlying neural populations. However, it is poorly understood what the exact mechanisms are that modulate the activity of neural populations as a whole and why the responses are so diverse in stimulation experiments. Despite this, electrical stimulation techniques are being developed for the treatment of neurological diseases in humans. To better understand these interactions, it is often necessary to simulate and analyze very large networks of neurons, which can be

**Data Availability Statement:** An implementation of the mean-field model is available as a Python library in our GitHub repository https://github.com/neurolib-dev/neurolib. The Python code of the comparison to AdEx network simulations, the

stimulation experiments, as well as the data analysis and the ability to reproduce all presented figures in this paper can be found at https://github.com/caglarcakan/stimulus_neural_populations.

**Funding:** This work was supported by the Deutsche Forschungsgemeinschaft (DFG, German Research Foundation) – Project number 327654276 – SFB 1315 (CC, KO) and the Research Training Group GRK1589/2 (CC). The funders had no role in study design, data collection and analysis, decision to publish, or preparation of the manuscript.

**Competing interests:** The authors have declared that no competing interests exist.

computationally demanding. In this theoretical paper, we present a reduced model of coupled neural populations that represents a piece of cortical tissue. This efficient model retains the dynamical properties of the large network of neurons it is based on while being several orders of magnitude faster to simulate. Due to the biophysical properties of the neuron model, an electric field can be coupled to the population. We show that weak electric fields often used in stimulation experiments can lead to entrainment of neural oscillations on the population level, and argue that the responses critically depend on the dynamical state of the neural system.

## Introduction

A paradigm which has proven to be successful in physical sciences is to systematically perturb a system in order to uncover its dynamical properties. This has also worked well for the different scales at which neural systems are studied. Mapping input responses experimentally has been key in uncovering the dynamical repertoire of single neurons [1, 2] and large neural populations such as *in vitro* cortical slice preparations [3]. It has been repeatedly shown that non-invasive *in vivo* brain stimulation techniques such as transcranial alternating current stimulation (tACS) can modulate oscillations of ongoing brain activity [4–6] and brain function [7, 8] and have enabled new ways for the treatment of clinical disorders such as epilepsy [9] or for enhancing memory consolidation during sleep [10]. Moreover, electrical input to neural populations can also originate from the active neural tissue itself, causing endogenous (intrinsic) extracellular electric fields which can modulate neural activity [11, 12].

However, a complete understanding of how electrical stimulation affects large networks of neurons remains elusive. For this reason, we present a computational framework for studying the interactions of time-varying electric inputs with the dynamics of large neural populations. Unlike *in vivo* and *in vitro* experimental setups, *in silico* models of electrical stimulation offer the possibility of studying a wide range of neuronal and stimulation parameters and might help to interpret experimental results.

For analytical tractability, theoretical research of the effects of electrical stimulation has relied on the use of mean-field methods to derive low-dimensional neural mass models [13–16]. Instead of simulating a large number of neurons, these models aim to approximate the population dynamics of interconnected neurons by means of dimensionality reduction. At the cost of disregarding the dynamics of individual neurons, it is possible to make statistical assumptions about large random neural networks and approximate their macroscopic behavior, such as the mean firing rate of the population.

Analyzing the state space of mean-field models has helped to characterize the dynamical states of coupled neural populations [17, 18]. Due to their computational efficiency, mean-field neural mass models are also typically used in whole-brain network models [19, 20], where they represent individual brain areas. This has made it possible to study the effects of external electrical stimulation on the ongoing activity of the human brain on a system level [21, 22].

Naturally, neural population models have to strike a balance between analytical tractability, computational cost, and biophysical realism. Thus, relating predictions from mean-field models to networks of biophysically realistic spiking neural populations is a challenging task. In order to bridge this gap, we present a mean-field population model based on a linear-nonlinear cascade [23, 24] of a large network of spiking adaptive exponential integrate-and-fire (AdEx or aEIF) neurons [25]. The AdEx neuron model in particular quite successfully reproduces the sub- and supra-threshold voltage traces of single pyramidal neurons found in cerebral cortex

**Fig 1. Schematic of the cortical motif.** Coupled populations of excitatory (red) and inhibitory (blue) neurons. **(a)** Mean-field neural mass model with synaptic feedforward and feedback connections. Each node represents a population. **(b)** Schematic of the corresponding spiking AdEx neuron network with connections between and within both populations. Both populations receive independent input currents with a mean $\mu_\alpha^{\text{ext}}$ and a standard deviation $\sigma_\alpha^{\text{ext}}$ across all neurons of population $\alpha \in \{E, I\}$.

[26, 27] while offering the advantage of having interpretable biophysical parameters. In our neural mass model, the set of parameters that determine its state space is the same set of parameters of the corresponding spiking neural network. This offers a straightforward way to relate the population dynamics as predicted by the mean-field model, including the effects of electrical inputs, to the biophysical parameters of the AdEx neuron.

In the following, we consider a classical motif of two delay-coupled populations of excitatory and inhibitory neurons that represents a cortical neural mass (Fig 1). We explore the rich dynamical landscape of this generic setup and investigate the effects of slow somatic adaptation currents on the population dynamics. We then apply time-varying electrical input currents to the excitatory population and observe frequency- and amplitude-dependent effects of the interactions between the stimulus and the endogenous oscillations of the system. We estimate the equivalent *extracellular electric field* amplitudes corresponding to these effects using previous results [28] of a spatially extended neuron model with a passive dendritic cable.

The main goals of this paper are thus twofold: Our main theoretical goal is to assess the validity of the mean-field approach in a wide range of parameters and with non-stationary electrical inputs in order to extend its validity to a more realistic case. Building on this, our second objective is to estimate realistic external field strengths at which experimentally relevant field effects are observed, such as attractor switching, frequency entrainment, and phase locking.

Predictions from mean-field theory are validated using simulations of large spiking neural networks. A close relationship of the mean-field model to the ground-truth model is established, proving its practical and theoretical utility. The mean-field model retains all dynamical states of the large network of individual neurons and predicts the interaction of the system with external electrical stimulation to a remarkable degree.

Our results confirm that weak fields with field strengths in the order of 1 V/m that are typically applied in tACS experiments can phase lock the population activity to the stimulus. Slightly stronger fields can entrain oscillation frequencies and induce population state switching. This also reinforces the notion that endogenous fields, generated by the activity of the brain itself, are expected to have a considerable effect on neural oscillations.

We believe that our results can help to understand the rich and plentiful observations in real neural systems subject to external stimulation and may provide a useful tool for studying the effects of electric fields on the population activity.

## Results

### The cortical mass model

We consider a cortical mass model which consists of two populations of excitatory adaptive (E) and inhibitory (I) exponential integrate-and-fire (AdEx) neurons (Fig 1). Both populations are delay-coupled and the excitatory population has a somatic adaptation feedback mechanism. The low-dimensional mean-field model (Fig 1a) is derived from a large network of spiking AdEx neurons (Fig 1b).

For the construction of the mean-field model, a set of conditions need to be fulfilled: We assume the number of neurons to be very large, all neurons within a population to have equal properties, and the connectivity between neurons to be sparse and random. Additional assumptions about the mathematical nature and a detailed derivation of the mean-field model is presented in the Methods section.

### Bifurcation diagrams: A map of the dynamical landscape

The E-I motif shown in Fig 1 can occupy various network states, depending on the baseline inputs to both populations. By gradually changing the inputs, we map out the state space of the system, depicted in the bifurcation diagrams in Fig 2. Small changes of the parameters of a nonlinear system can cause sudden and dramatic changes of its overall behavior, called bifurcations. Bifurcations separate the state space into distinct regions of network states between which the system can transition from one to another. In our case, the dynamical state of the E-I system depends on external inputs to both subpopulations, which are directly affected by external electrical stimulation and other driving sources, e.g. inputs from other neural populations such as other brain regions.

Comparing the bifurcation diagrams of the mean-field model (Fig 2a and 2c) to the ground truth spiking AdEx network (Fig 2b and 2d) demonstrates the similarity between both dynamical landscapes. Transitions between states take place at comparable baseline input values and in a well-preserved order.

Since the space of possible biophysical parameter configurations is vast, we focus on two variants of the model: one without a somatic adaptation mechanism, Fig 2a and 2b, and one with finite sub-threshold and spike-triggered adaptation in Fig 2c and 2d. Both variants feature distinct states and dynamics.

**Bistable up- and down-states without adaptation.**    Fig 2a and 2b show the bifurcation diagrams of the E-I system without somatic adaptation. There are two stable fixed-point solutions of the system with a constant firing rate: a low-activity *down-state* and a high-activity *up-state*. These macroscopic states correspond to asynchronous irregular firing activity on a microscopic level [13] (see S1 Fig). In accordance with previous studies [29–31], at larger mean background input currents, there is a *bistable* region in which the *up-state* and the *down-state* coexist. At smaller mean input values, the recurrent coupling of excitatory and inhibitory neurons gives rise to an oscillatory limit cycle $LC_{EI}$ with an alternating activity between the two populations. Example time series of the population rates of E and I inside the limit cycle are shown in Fig 2e and 2f (top row). The frequency inside the oscillatory region depends on the inputs to both populations and ranges from 8 Hz to 29 Hz in the mean-field model and from 4 Hz to 44 Hz in the AdEx network for the parameters given (see S4 Fig).

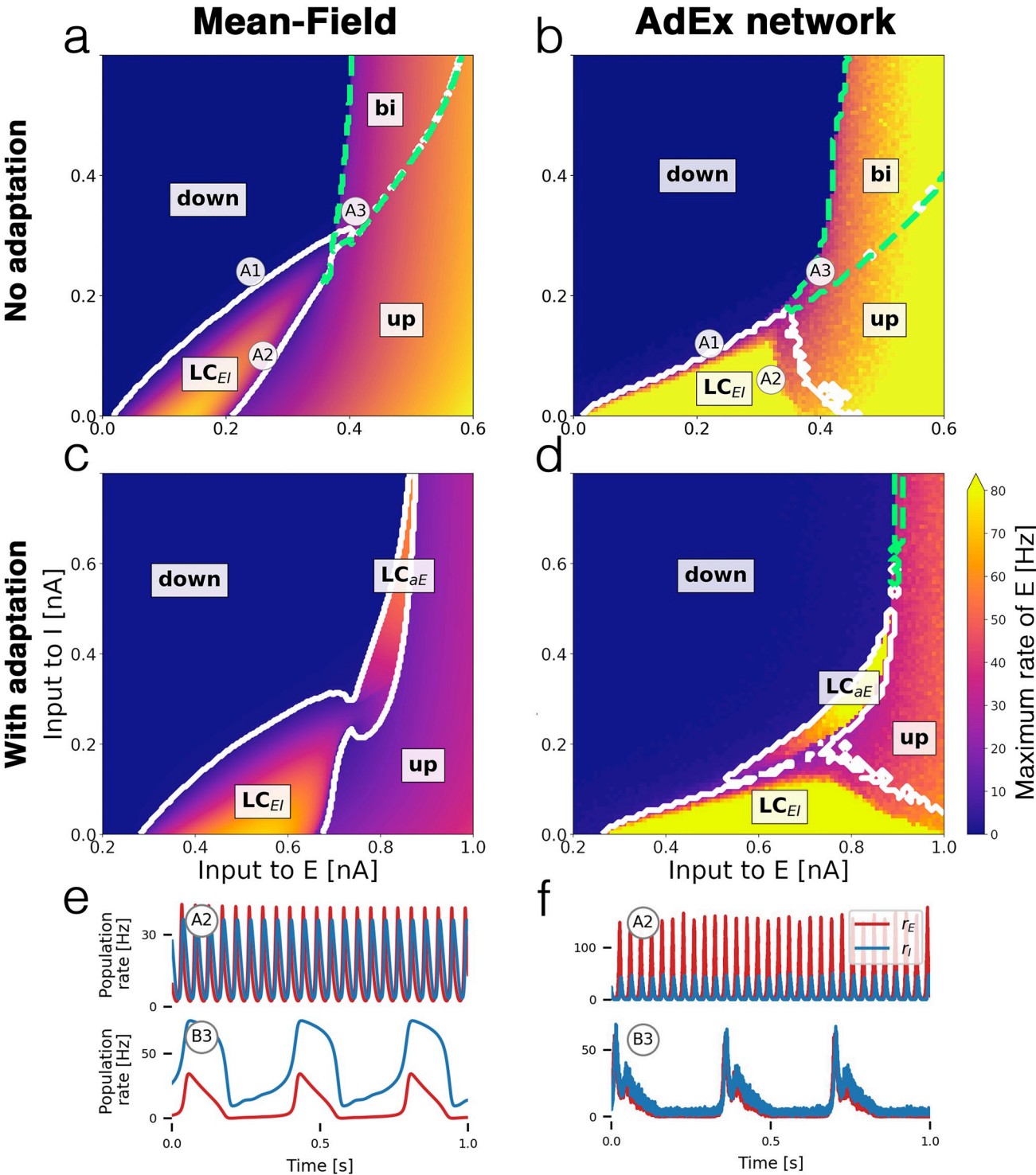

**Fig 2. Bifurcation diagrams and time series.** Bifurcation diagrams depict the state space of the E-I system in terms of the mean external input currents $C \cdot \mu_\alpha^{\text{ext}}$ to both subpopulations $\alpha \in \{E, I\}$. **(a)** Bifurcation diagram of mean-field model without adaptation with *up* and *down-states*, a bistable region *bi* (green dashed contour) and an oscillatory region $LC_{EI}$ (white solid contour). **(b)** Diagram of the corresponding AdEx network with $N = 50 \times 10^3$ neurons. **(c)** Mean-field model with somatic adaptation. The bistable region is replaced by a slow oscillatory region $LC_{aE}$. **(d)** Diagram of the corresponding AdEx network. The color in panels a—d indicate the maximum population rate of the excitatory population (clipped at 80 Hz). **(e)** Example time series of the population rates of excitatory (red) and inhibitory (blue) populations at point A2 (top row) which is located in the fast excitatory-inhibitory limit cycle $LC_{EI}$, and at point B3 (bottom row) which is located in the slow limit cycle $LC_{aE}$. **(f)** Time series at corresponding points for the AdEx network. All parameters are listed in Table 1. The mean input currents to the points of interest A1-A3 and B3-B4 are provided in Table 2.

**Table 1. Summary of the model parameters.** Parameters apply for the Mean-Field model and the spiking AdEx network.

| Parameter | Value | Description |
|---|---:|---|
| $\sigma^{\text{ext}}$ | 1.5 mV/$\sqrt{\text{ms}}$ | Standard deviation of external input |
| $K_e$ | 800 | Number of excitatory inputs per neuron |
| $K_i$ | 200 | Number of inhibitory inputs per neuron |
| $c_{EE}, c_{IE}$ | 0.3 mV/ms | Maximum AMPA PSC amplitude [67] |
| $c_{EI}, c_{II}$ | 0.5 mV/ms | Maximum GABA PSC amplitude, [67] |
| $J_{EE}$ | 2.4 mV/ms | Maximum synaptic current from E to E |
| $J_{IE}$ | 2.6 mV/ms | Maximum synaptic current from E to I |
| $J_{EI}$ | −3.3 mV/ms | Maximum synaptic current from I to E |
| $J_{II}$ | −1.6 mV/ms | Maximum synaptic current from I to I |
| $\tau_{s,E}$ | 2 ms | Excitatory synaptic time constant |
| $\tau_{s,I}$ | 5 ms | Inhibitory synaptic time constant |
| $d_E$ | 4 ms | Synaptic delay to excitatory neurons |
| $d_I$ | 2 ms | Synaptic delay to inhibitory neurons |
| $C$ | 200 pF | Membrane capacitance |
| $g_L$ | 10 nS | Leak conductance |
| $\tau_m$ | $C/g_L$ | Membrane time constant |
| $E_L$ | −65 mV | Leak reversal potential |
| $\Delta_T$ | 1.5 mV | Threshold slope factor |
| $V_T$ | −50 mV | Threshold voltage |
| $V_s$ | −40 mV | Spike voltage threshold |
| $V_r$ | −70 mV | Reset voltage |
| $T_{\text{ref}}$ | 1.5 ms | Refractory time |
| $a$ | 15 nS | Subthreshold adaptation conductance |
| $b$ | 40 pA | Spike-triggered adaptation increment |
| $E_A$ | −80 mV | Adaptation reversal potential |
| $\tau_A$ | 200 ms | Adaptation time constant |

All macroscopic network states of the AdEx network are represented in the mean-field model. The bifurcation line that marks the transition from the *down-state* to LC$_{\text{EI}}$ appears at a similar location in the state space, close to the diagonal at which the mean inputs to E and I are equal, in both, the mean-field and the spiking network model. However, the shape and width of the oscillatory region, as well as the amplitudes and frequencies of the oscillations differ. In Fig 2e and 2f (top row), the differences are due to the location of the chosen points A2 in the bifurcation diagrams, which are not particularly chosen to precisely match each other in

**Table 2. Values of the mean external inputs to the excitatory ($\mu_E^{\text{ext}}$) and the inhibitory population ($\mu_I^{\text{ext}}$) in units of nA for points of interest in the bifurcation diagrams Fig 2.**

| Point | Mean-Field model | | AdEx network | | Dynamical state |
|---|---|---|---|---|---|
| | $C \cdot \mu_E^{\text{ext}}$ | $C \cdot \mu_I^{\text{ext}}$ | $C \cdot \mu_E^{\text{ext}}$ | $C \cdot \mu_I^{\text{ext}}$ | |
| A1 | 0.24 | 0.24 | 0.22 | 0.12 | down |
| A2 | 0.26 | 0.1 | 0.32 | 0.3 | LC$_{\text{EI}}$ |
| A3 | 0.41 | 0.34 | 0.4 | 0.24 | bi |
| B3 | 0.8 | 0.36 | 0.76 | 0.24 | LC$_{\text{aE}}$ |
| B4 | 0.76 | 0.4 | 0.68 | 0.24 | down |

amplitude or frequency but rather in the approximate location in the state space. Overall, the AdEx network exhibits larger amplitudes across the oscillatory regime (S2 Fig) and the excitatory amplitudes are larger than the inhibitory amplitudes (S3 Fig). Another notable difference is the small bistable overlap of the *up-state* region with the oscillatory region $LC_{EI}$ in the mean-field model (Fig 2a) which could not be observed in the AdEx network.

**Somatic adaptation causes slow oscillations.**   In Fig 2c and 2d, bifurcation diagrams of the system with somatic adaptation are shown. Compared to Fig 2a and 2b (without adaptation), the state space, including the oscillatory region $LC_{EI}$, is shifted to the right, meaning that larger excitatory input currents are necessary to compensate for the inhibiting sub-threshold adaptation currents. The most notable effect that is caused by adaptation is the appearance of a slow oscillatory region labeled $LC_{aE}$ in Fig 2c and 2d. The reason for the emergence of this oscillation is the destabilizing effect the inhibiting adaptation currents have on the *up-state* inside the *bistable* region [29–31]. As the mean adaptation current builds up due to a high population firing rate, the *up-state* "decays" and the system transitions to the *down-state*. The resulting low activity causes a decrease of the adaptation currents which in turn allow the activity to increase back to the *up-state*, resulting in a slow oscillation. These low-frequency oscillations range from 0.5 Hz to 5 Hz for the parameters given.

The bifurcation diagrams in Fig 3 show how the emergence of the slow oscillation depends on the adaptation mechanism. Increasing the subthreshold adaptation parameter primarily shifts the state space to the right whereas a larger spike-triggered adaptation parameter value enlarges oscillatory regions. Both parameters cause the *bistable* region to shrink until it is eventually replaced by a slow oscillatory region $LC_{aE}$. Again, the state space of the mean-field model (Fig 3a) reflects the AdEx network (Fig 3b) accurately.

## Time-varying stimulation and electric field effects

To describe the time-dependent properties of the system, we study the effects of time-varying external stimulation and the interactions with ongoing oscillatory states. External stimulation is implemented by coupling an electric input current to the excitatory population. This additional input current may be a result of an externally applied electric field or synaptic input from other neural populations. For the cases without adaptation, we can calculate an equivalent extracellular electric field strength that correspond to the effects of an input current (see Methods).

Since due to the presence of apical dendrites, excitatory neurons in the neocortex are most susceptible to electric fields [32], only time-varying input to the excitatory population is considered. This choice is also motivated in the context of inter-areal brain network models where connections between brain areas are usually considered between excitatory subpopulations.

Given the multitude of possible states of the system, its response to external input critically depends on the dynamical landscape around its current state. It is important to keep in mind that the bifurcation diagrams (Figs 2 and 3) are valid only for constant external input currents. However, they provide a helpful estimation of the dynamics of the non-stationary system assuming that the bifurcation diagrams do not change too much as we vary the input parameter $\mu_e^{ext}(t)$ over time.

Fig 4a–4f show how a step current input pushes the system in and out of specific states of the E-I system. A positive step current represents a movement in the positive direction of the $\mu_e^{ext}$-axis in Fig 2. Fig 4a and 4b show input-driven transitions from the low-activity *down-state* to the fast oscillatory limit cycle $LC_{EI}$. Similar behavior can be observed in Fig 4c and 4d where we push the system's state from $LC_{EI}$ to the *up-state*, effectively being able to turn oscillations on and off with a direct input current.

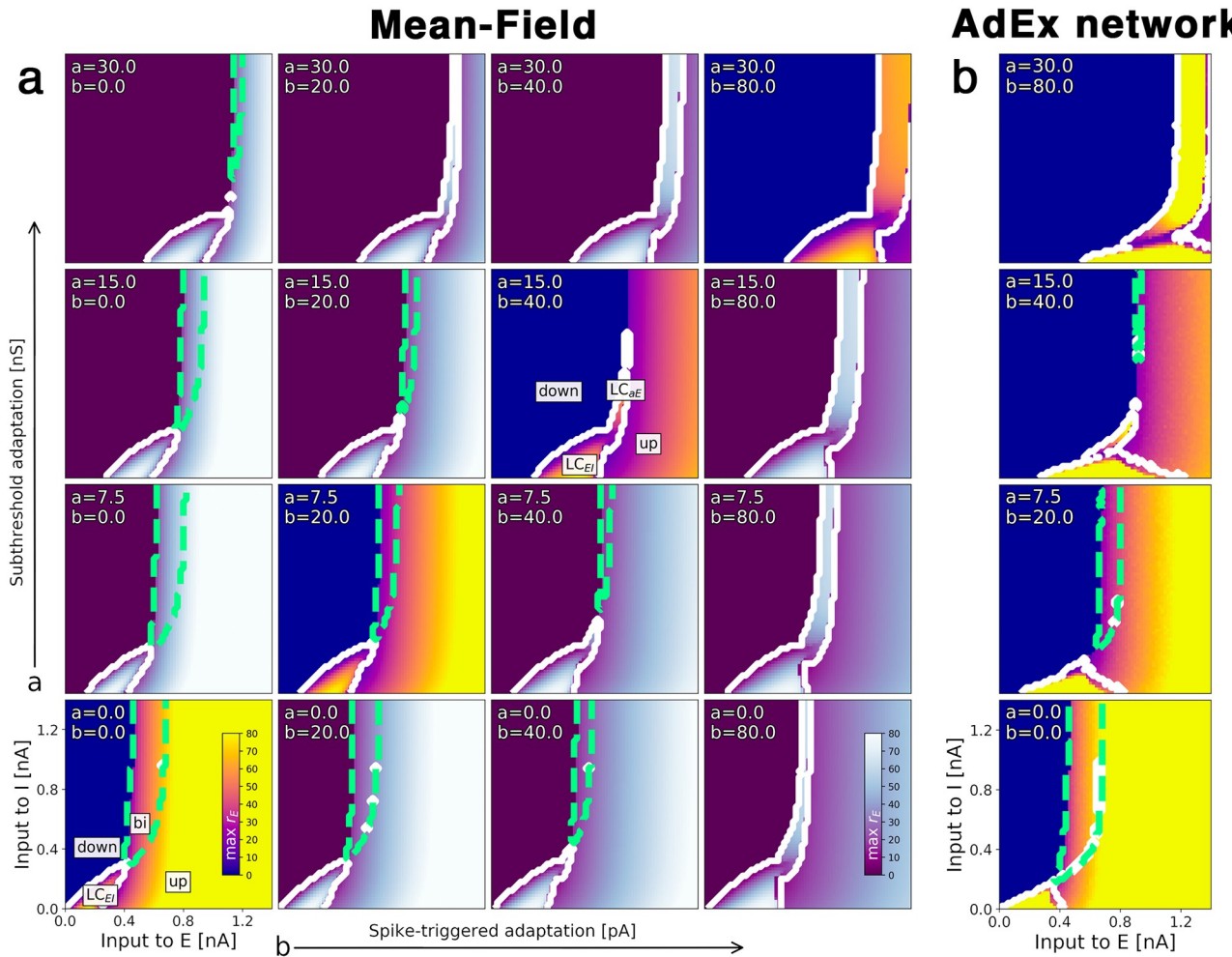

**Fig 3. Transition from multistability to slow oscillation is caused by somatic adaptation.** Bifurcation diagrams depending on the external input currents $C \cdot \mu_\alpha^{\text{ext}}$ to both populations $\alpha \in \{E, I\}$ for varying somatic adaptation parameters $a$ and $b$. The color indicates the maximum rate of the excitatory population. Oscillatory regions have a white contour, bistable regions have a green dashed contour. **(a)** Bifurcation diagrams of the mean-field model. On the diagonal (bright-colored diagrams), adaptation parameters coincide with (b). **(b)** Bifurcation diagrams of the corresponding AdEx network, N = $20 \times 10^3$. All parameters are listed in Table 1.

Inside the *bistable* region, we can use the hysteresis effect to transition between the *down-state* and the *up-state* and vice versa. After application of an initial push in the desired direction, the system remains in that state, reflecting the system's bistable nature.

With adaptation turned on, a slow oscillatory input current can entrain the ongoing oscillation. In Fig 4g and 4h, the oscillation is initially out of phase with the external input but is quickly phase-locked. Placed close to the boundary of the slow oscillatory region $LC_{aE}$, we show in Fig 4i and 4j how an oscillatory input with a similar frequency as the limit cycle periodically drives the system from the *down-state* into one oscillation period.

Close inspection of Fig 4b shows that the fast oscillation in the AdEx network has a varying amplitude, in contrast to the mean-field model Fig 4a. This difference is due to noise resulting from the finite size of the AdEx network and decreases as the network size *N* increases. (see S8 Fig) All of the state transitions take longer for the AdEx network, as it is visible in Fig 4d for example. Additionally, transitions to the *up-state* and the slow limit cycle $LC_{aE}$ are

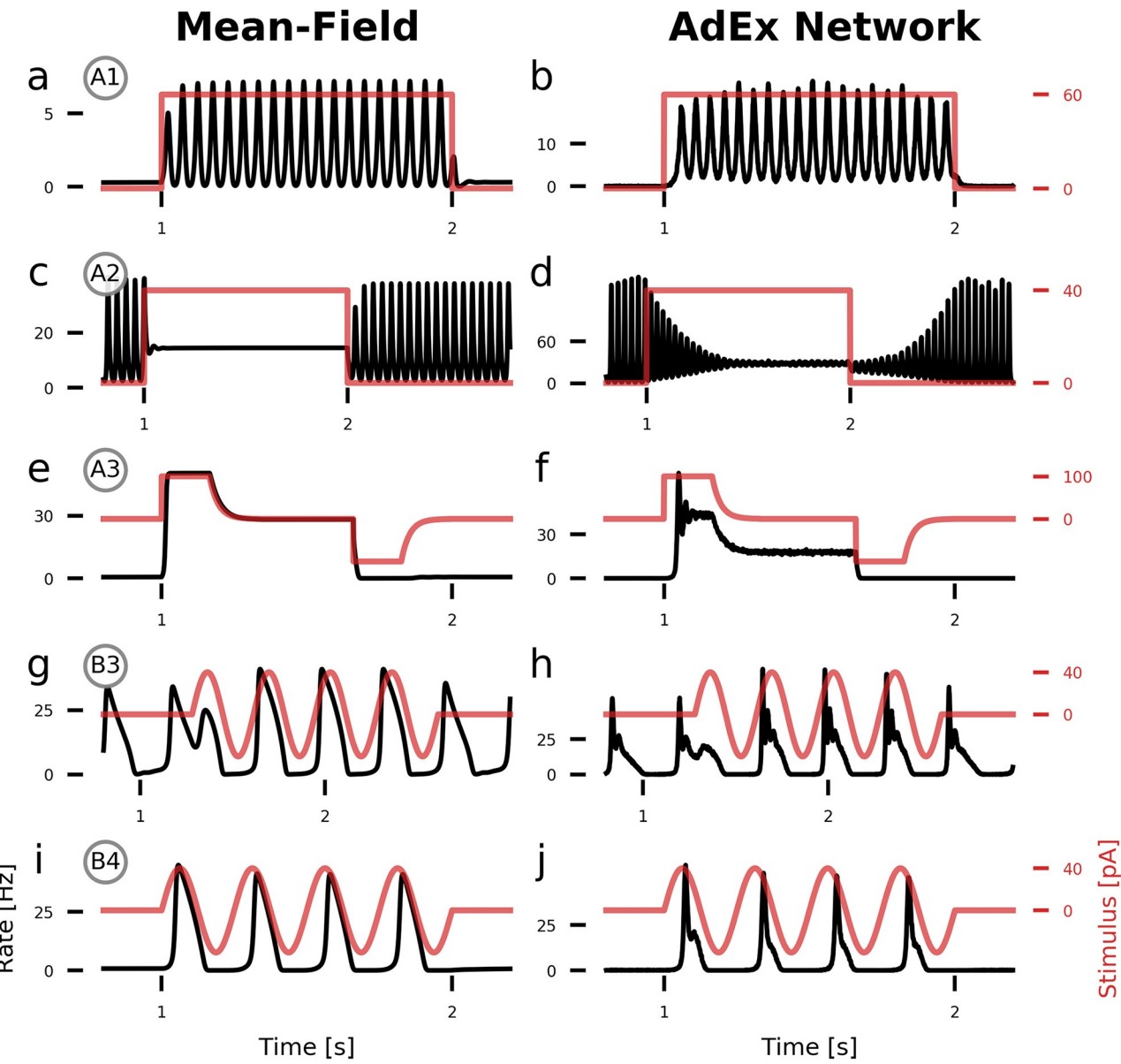

**Fig 4. State-dependent population response to time-varying input currents.** Population rates of the excitatory population (black) with an additional external electrical stimulus (red) applied to the excitatory population. (**a, b**) A DC step input with an amplitude of 60 pA (equivalent E-field amplitude: 12 V/m) pushes the system from the low-activity fixed point into the fast limit cycle $LC_{EI}$. (**c, d**) A step input with amplitude 40 pA (8 V/m) pushes the system from $LC_{EI}$ into the *up-state*. (**e, f**) In the multistable region *bi*, a step input with amplitude 100 pA (20 V/m) pushes the system from the *down-state* into the *up-state* and back. (**g, h**) Inside the slow oscillatory region $LC_{aE}$, an oscillating input current with amplitude 40 pA and a (matched) frequency of 3 phase-locks the ongoing oscillation. (**i, j**) A slow 4 Hz oscillatory input with amplitude 40 pA drives oscillations if the system is close to the oscillatory region $LC_{aE}$. For the AdEx network, $N = 100 \times 10^3$. All parameters are given in Table 1. The parameters of the points of interest are given in Table 2.

accompanied by transient ringing activity (Fig 4f and 4h), which is not well-captured by the mean-field model.

**Frequency entrainment with oscillatory input.** To study the frequency-dependent response of the E-I system, we vary the amplitude and frequency of an oscillatory input to the excitatory population (Fig 5). The unperturbed system is parameterized to be in the fast limit cycle $LC_{EI}$ with its endogenous frequency $f_0$.

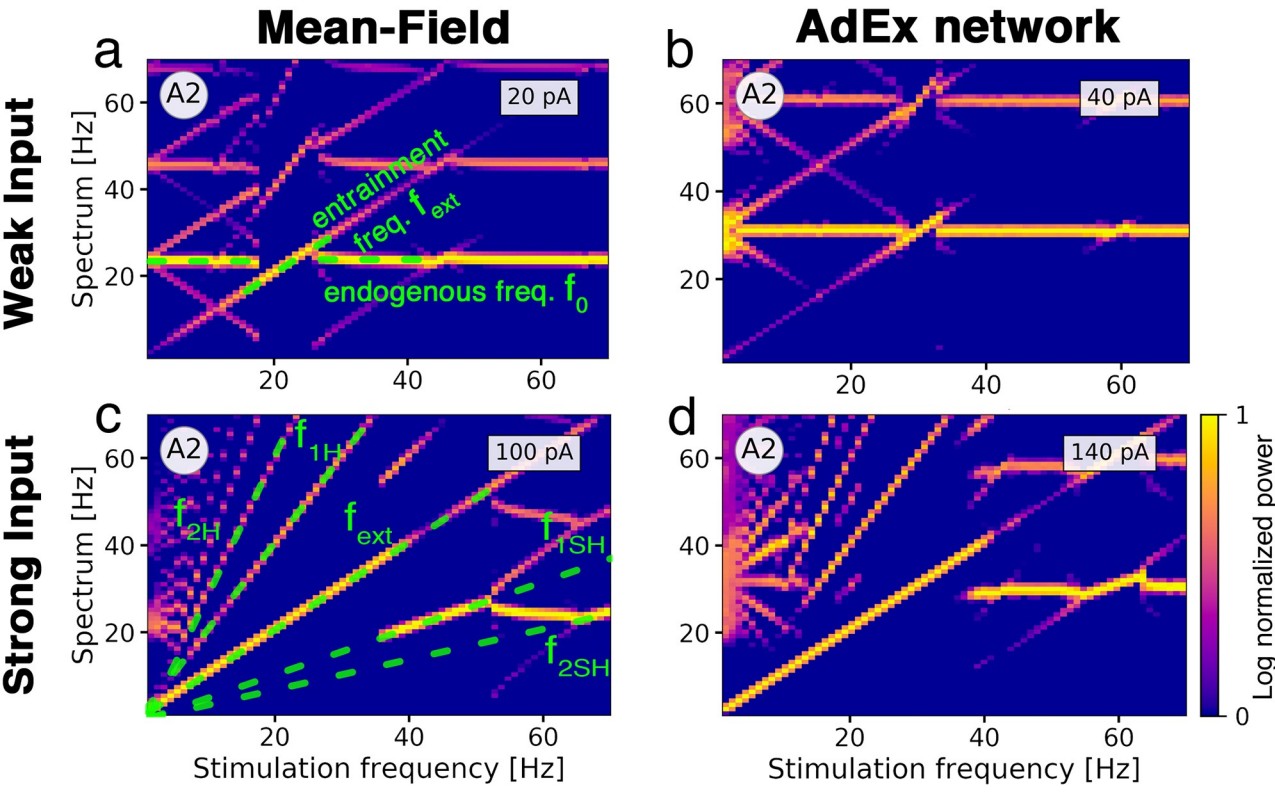

**Fig 5. Frequency entrainment of the population activity in response to oscillatory input.** The color represents the log-normalized power of the excitatory population's rate frequency spectrum with high power in bright yellow and low power in dark purple. (**a**) Spectrum of the mean-field model parameterized at point A2 with an ongoing oscillation frequency of $f_0 = 22$ Hz (horizontal green dashed line) in response to a stimulus with increasing frequency and an amplitude of 20 pA. An external electric field with a resonant stimulation frequency of $f_0$ has an equivalent strength of 1.5 V/m. The stimulus entrains the oscillation from 18 Hz to 26 Hz, represented by a dashed green diagonal line. At 27 Hz, the oscillation falls back to its original frequency $f_0$. At a stimulation frequency of $2f_0$, the ongoing oscillation at $f_0$ locks again to the stimulus in a smaller range from 43 Hz to 47 Hz. (**b**) AdEx network with $f_0 = 30$ Hz. Entrainment with an input current of 40 pA is effective from 27 Hz to 33 Hz. Electric field amplitude with frequency $f_0$ corresponds to 2.5 V/m. (**c**) Mean-field model with a stimulus amplitude of 100 pA (7.5 V/m at 22 Hz). Green dashed lines mark the driving frequency $f_{ext}$ and its first and second harmonics $f_{1H}$ and $f_{2H}$ and subharmonics $f_{1SH}$ and $f_{2SH}$. Entrainment is effective from the lowest stimulation frequency up to 36 Hz at which the oscillation falls back to a frequency of 20 Hz. New diagonal lines appear due to interactions of the endogenous oscillation with the entrained harmonics and subharmonics. (**d**) AdEx network with stimulation amplitude of 140 pA (8.75 V/m at 30 Hz). For the AdEx network, $N = 20 \times 10^3$. All parameters are given in Tables 1 and 2.

The external stimulus with frequency $f_{ext}$ entrains the ongoing oscillation in a range around $f_0$, the resonant frequency of the system. Here, the ongoing oscillation essentially follows the external drive and adjusts its frequency to it (Fig 5a). A second (narrower) range of frequency entrainment appears as $f_{ext}$ approaches $2f_0$, representing the ability of the input to entrain oscillations at half of its frequency. Due to interference of the frequencies of ongoing and external oscillations, the spectrum has peaks at the difference of both frequencies which appear as X-shaped patterns in the frequency diagrams. The AdEx network shows a similar behavior (Fig 5b), albeit the range of entrainment is smaller than in the mean-field model, despite the stimulation amplitude being twice as large.

For stronger oscillatory input currents, the range of frequency entrainment is widened considerably. In Fig 5c and 5d, the input dominates the spectrum at very low frequencies. The peak of the spectrum reverts back to approximately $f_0$ if the external frequency $f_{ext}$ is close to the first harmonic $2f_0$ of the endogenous frequency. We see multiple lines emerging in the frequency spectra which correspond to the harmonics and subharmonics of the external

frequency and its interaction with the endogenous frequency $f_0$, creating complex patterns in the diagrams. Differences between the spectrograms of the AdEx network in Fig 5d and the mean-field model Fig 5c can be largely attributed to the fact that the AdEx network consistently needs stronger inputs to obtain the same effect as in the mean-field model. This results in horizontal lines in areas where frequency entrainment is not effective and in faint and short diagonal lines between the lines that represent the (sub-)harmonics which are caused by interactions with the (sub-)harmonics. In the mean-field model, we mainly observe clear diagonal lines, indicating successful entrainment. Another source for the differences is the inherently noisy dynamics of the AdEx network, due to its finite size (see S8 Fig).

Overall, there is a good qualitative agreement of the frequency spectra of both models, reflecting that interactions of time-varying external inputs and ongoing oscillations are well-represented by the mean-field model.

**Phase locking with oscillatory input.** Here we quantify the ability of an oscillating external input current to the excitatory population to synchronize an ongoing neural oscillation to itself if both frequencies, the driver and the endogenous frequency, are close to each other (frequency matching). An example time series of a stimulus entraining an ongoing slow oscillation is shown in Fig 4h.

In Fig 6, we find phase locking by measuring the time course of the phase difference between the stimulus and the population rate. If phase locking is successful, the phase difference remains constant. In Fig 6a, the region of phase locking for an external input of frequency $f_{ext}$ is centered around the endogenous frequency $f_0$ of the unperturbed system. Increasing the stimulus amplitude widens the range around $f_0$ at which phase locking is effective, producing Arnold tongues in the diagram. An example time series of successful phase locking inside this region is shown in Fig 6c at point 1. If the input is not able to phase-lock the ongoing activity, a small difference between the driver frequency $f_{ext}$ and $f_0$ can cause a slow beating of the activity with a frequency of roughly the difference $|f_{ext} - f_0|$. Thus, a small frequency mismatch can produce a very slowly oscillating activity (Fig 6c at points 2-4). Fig 6d at point 2 shows the same drifting effect in the AdEx network. Due to finite-size noise in the AdEx network, an irregular switching between synchrony and asynchrony can be observed at the edges of the phase locking region in Fig 6d at point 3. Compared to the mean-field model, the frequency of the beating activity in the AdEx network is less regular (Fig 6d at point 4).

In the phase locking diagrams Fig 6a and 6b, the equivalent external electric field amplitudes are shown. Small amplitudes (0.2 V/m for the mean-field model, 0.5 V/m for the AdEx network) are able to phase-lock the ongoing oscillations if the frequencies roughly match.

## Discussion

In this paper, we explored the dynamical properties of a cortical neural mass model of excitatory and inhibitory (E-I) adaptive exponential integrate-and-fire (AdEx) neurons by studying their response to external electrical stimulation. Results from a low-dimensional mean-field model of a spiking AdEx neuron network were compared with large network simulations (Fig 1). The mean-field model provides an accurate and computationally efficient approximation of the mean population activity and the mean membrane potentials of the AdEx network if all neurons are equal, the number of neurons is large, and the connectivity is sparse and random. The mean-field model and the AdEx network share the same set of biophysical parameters (see Methods). The biophysical parameters of the AdEx neuron allow us to model realistic external electric currents and extracellular field strengths [28]. In the mean-field description, this allows us to model stimulation to the whole population in various network states.

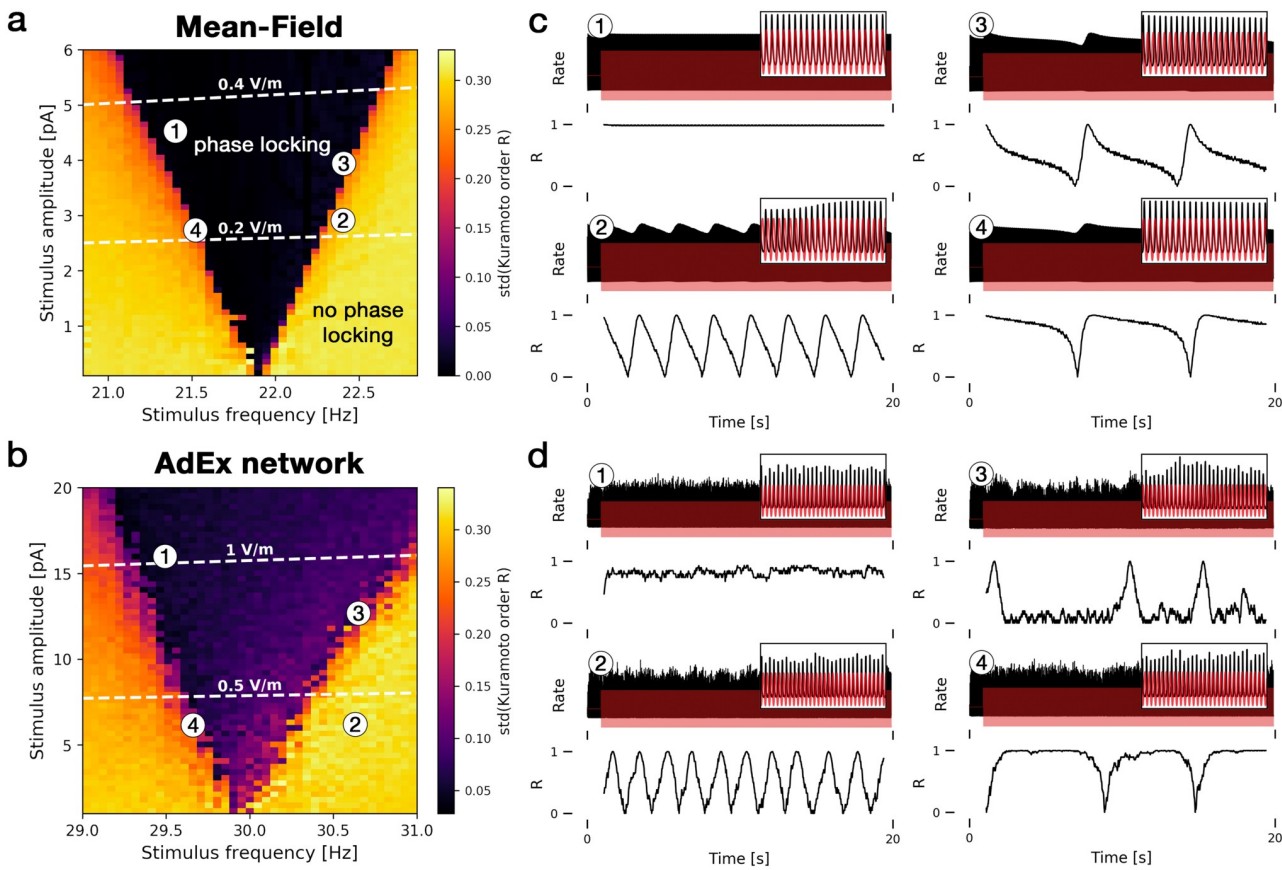

**Fig 6. Phase locking of ongoing oscillations via weak oscillatory inputs.** The left panels show heatmaps of the level of phase locking for **(a)** the mean-field model and **(b)** the AdEx network for different stimulus frequencies and amplitudes. Dark areas represent effective phase locking and bright yellow areas represent no phase locking. Phase locking is measured by the standard deviation of the Kuramoto order parameter $R(t)$ which is a measure for phase synchrony. White dashed lines correspond to electric fields with equivalent strengths in V/m. **(c)** Time series of four points indicated in (a) with the excitatory population's rate in black and the external input in red (upper panels). In the lower panels, the Kuramoto order parameter $R(t)$ is shown, measuring the phase synchrony between the population rate and the external input. Constant $R(t)$ represents effective phase locking (phase difference between rate and input is constant), fluctuating $R(t)$ indicates dephasing of both signals, hence no phase locking. **(d)** Corresponding time series of points in (b). Both models are parameterized to be in point A2 inside the fast oscillatory region $LC_{EI}$. Insets show zoomed-in traces from 15 to 16 seconds. For the AdEx network, $N = 20 \times 10^3$. All parameters are given in Table 1.

Bifurcation diagrams (Fig 2) provide a map of the possible states as a function of the external inputs to both, excitatory and inhibitory, populations. A comparison of the diagrams of the mean-field model to the corresponding AdEx network model reveals a high degree of similarity of the state spaces. Each attractor of the AdEx network is represented in the mean-field model in a one-to-one fashion which allows for accurate predictions of the state of the spiking neural network using the low-dimensional mean-field model.

We have focused our attention on bifurcations caused by changes of the mean external input currents which can represent background inputs to the neural population or electrical inputs from external stimulation. It is worth noting that other parameters, such as coupling strengths and adaptation parameters, can cause bifurcations as well. Overall, the specific shape of the dynamical landscape depends on numerous parameters. However, extensive parameter explorations indicated that the accuracy of the mean-field model as well as the overall structure of the bifurcation diagrams presented in this paper was fairly robust to changes of the coupling strengths and therefore representative for this E-I system (see S5 and S6 Figs).

Without a somatic adaptation feedback mechanism, the population rate can occupy four distinct states: a *down-state* with very low activity, an *up-state* with constant high activity representing an asynchronous firing state of the neurons (see S1 Fig), a *bistable* regime where *down-state* and *up-state* coexist and an oscillatory state where the activity alternates between the excitatory and the inhibitory population at a low gamma frequency.

## Somatic adaptation causes slow network oscillations

The AdEx neuron model allows for incorporation of a slow potassium-mediated adaptation current, typically found in cortical pyramidal neurons [33]. Due to somatic adaptation, in the bistable region, the *up-state* loses its stability. The bistable region transforms into a second oscillatory regime (Fig 3) in which the population activity oscillates at low frequencies between 0.5 Hz and 5 Hz. This oscillatory region coexists with the fast excitatory-inhibitory oscillation. Other computational studies have focused on the origin of this adaptation-mediated oscillation [29–31], the interaction of adaptation with noise-induced state switching between *up-* and *down-state* [29, 34, 35] and how adaptation affects the intrinsic timescales of the network [36, 37].

## Electric field effects and relation to experimental observations

Using the bifurcation diagrams (Fig 2), we mapped out several points of interest that represent different network states. The type of reaction to external stimulation depends on the current state of the system, as seen in the population time series during stimulation in Fig 4. Close to edges of attractors, direct currents can cause bifurcations and trigger a sudden change of the dynamics, such as transitions from a low activity *down-state* to a state with oscillatory activity.

*In vitro* stimulation experiments with electric fields [3] have shown that (time-constant) direct fields are able to switch on and off oscillations at field strengths of 6 V/m. In Fig 4, we could observe this at 8 − 12 V/m, when the system if placed close to the oscillatory state. This difference in amplitudes can be attributed to the chosen initial state of the system and could be reduced if the background inputs were parameterized closer to the limit cycle.

Inside oscillatory regions, oscillatory input causes phase locking and frequency entrainment. Frequency entrainment is the ability of an external stimulus to force the endogenous oscillation to follow its frequency. To study how frequency entrainment depends on the frequency and amplitude of the stimulus, we analyzed frequency spectrograms of the population activity when subject to external oscillating stimuli with increasing frequencies (Fig 5). We observed shifts of the peak frequency around the endogenous frequency at amplitudes corresponding to field strengths of 1.5 V/m in the mean-field model and 2.5 V/m in the AdEx network. Similar effects have been reported in *in vitro* experiments, where the frequency of the ongoing oscillations changed along with the stimulus frequency [3, 38–40].

Interestingly, the field amplitudes at which frequency entrainment is visible are on the same order of endogenous fields in the brain, generated by the neural activity itself. Electrophysiological experiments show that these fields can be as strong as 3.5 V/m [11] and that ephaptic coupling plays a significant role in the brain [12]. Our findings support this observation from a theoretical perspective, since one of our main results is that considerable effects on the population dynamics are expected at these field strengths.

We also observed frequency entrainment of the subharmonics of the endogenous oscillation as it was shown in *in vitro* experiments in Refs. [3, 39] and its harmonics in Ref. [38]. This effect could be valuable for experimental conditions where it is impractical to use stimulation frequencies close to the endogenous frequency of ongoing oscillations in the studied neural

system. The range of frequency entrainment around the natural frequency of the endogenous oscillation widens as the stimulus amplitude increases, which was also observed in similar computational studies [41, 42].

If the stimulus frequency is close to the endogenous frequency (frequency matching), an oscillatory stimulus can force the ongoing oscillation to synchronize its phase with the stimulus, known as phase locking, phase entrainment, or coherence. Phase locking of ongoing brain activity to a stimulus has been observed in multiple noninvasive brain stimulation studies, including Refs. [43–45], and it has been shown to affect information processing properties of the brain [7, 8] as particularly sensory information processing depends on phase coherence of oscillations between distant brain regions [46, 47]. Compared to frequency entrainment, very weak input currents are able to phase lock ongoing oscillations. In agreement with these experiments, we find phase locking to be effective at electric field strengths of around 0.5 V/m (Fig 6), which is in the range of typical field strengths generated by transcranial alternating current stimulation (tACS) [48].

To summarize: Our results confirm the interesting notion that, while weak electric fields with strengths in the order of 1 V/m that are typically applied in tACS experiments have only a small effect on the membrane potential of a single neuron [32], the effects on the network however, and therefore on the dynamics of the population as a whole, can be quite significant, which was also observed experimentally [49]. This indicates that field effects are strongly amplified in the network. Considering slightly stronger fields, our results suggest that endogenous fields, generated by the activity of the brain itself, are expected to have a considerable effect on neural oscillations, facilitating phase and frequency synchronization across neighboring cortical brain areas.

## Validity of the mean-field method and limitations of our approach

We found all observed input-dependent effects in the AdEx network to be well-represented by the mean-field model, which demonstrates its accuracy also in the non-stationary case. However, partly due to the difference of parameters that define the states in the bifurcation diagrams, the AdEx network consistently requires larger input amplitudes in order to cause the same effect size as observed in the mean-field model (Figs 5 and 6). Although it was not investigated here, we hypothesize that the number of neurons might play an important role in how much external inputs are amplified within a network.

Related to this is the fact that our approximation assumes the number of neurons to be infinitely large. Therefore, differences between the mean-field model and the AdEx network in the case without external stimulation also depend on the network size. However, we find good agreement between the bifurcation diagrams for as low as $N = 4 \times 10^3$ neurons (S7 Fig). With increasing network size, the amplitudes of oscillations in the AdEx network shown in Fig 4b approach the predictions of the mean-field model and become less irregular (S8 Fig).

Comparing the bifurcation diagrams of both models (Fig 2), the shape of the oscillatory region as well as the frequencies of the oscillations differ (see S4 Fig). We suspect that the oscillatory states are where the steady-state approximations that are used to construct the mean-field model break down due to the fast temporal dynamics in this state. Hence, both models have notable differences between the oscillatory regions.

Sharp transitions between states cause transient effects that are visible as ringing oscillations in the population firing rate, typically observed in simulations of spiking networks (such as in Fig 4f and 4h) or experimentally [50]. The poor reproduction of these oscillations in our model is likely due to the use of an exponentially-decaying linear response function instead of a damped oscillation which would be a better approximation of the true response (c.f.

Ref. [23]). This constitutes a possible improvement of our work. Recent advancements in mean-field models of cortical networks [51] can account for its finite size as well as reproduce transient oscillations caused by sharp input onsets.

Another important limitation of our method is the assumption of homogeneous and weak synaptic coupling in the mathematical derivation of our mean-field model. Synapses in the brain are known to be log-normally distributed [52] with long tails, implying the existence of few but strong synapses. Other computational papers have specifically focused on the effect of strong synapses on the population activity (cf. [53] and [54]). Therein, the incorporation of strong synapses causes the emergence of a new asynchronous state in which the firing rates of individual neurons fluctuate strongly, similar to chaotic states studied in networks of rate neurons [55], which are qualitatively different from the *up-state* that we observed (see S1 Fig). In Ref. [53], the author shows that firing rate models similar to what we consider break down and cannot capture the large fluctuations present in this state. We therefore conclude that our mean-field model is limited to describing only weak synaptic coupling.

Furthermore, a number of assumptions were made in our model of electric field interaction. Most importantly, we have chosen typical morphological and electrophysiological parameters of a ball-and-stick neuron model to represent pyramidal cells in layer 4/5 of cortex (see Methods). Despite the simplicity of the ball-and-stick model, it was shown in Ref. [28] that it can reproduce the somatic polarization of a pyramidal cell in a weak and uniform field. This was then translated into effective input currents to point neurons which lack any morphological features. In addition to the crude assumption that all neurons have the same simple morphology (effects of a more complex morphology were studied in Ref. [56]), we also assumed perfect alignment of the dendritic cable to the external electric field. While the latter might be a good approximation for a local region of the cortex it is not the case for the brain as a whole with its folded structure. It has been shown that the somatic polarization of a neuron strongly depends on the angle between the neuron's main axis and the electric field [32].

We only focused on field effects that are caused by the dendrite. Although we expect that, in principle, axons could contribute to the somatic polarization in a weak and uniform electric field, their contribution could be relatively small, since most cortical axons are not geometrically aligned with each other the way that dendrites are organized in the columnar structure of the cortex [57].

Finally, we assumed that the field effects are only subthreshold such that our results do not generalize to stimulation scenarios with strong electric fields that can elicit action potentials by themselves.

## Conclusion

Overall, our observations confirm that a sophisticated mean-field model of a neural mass is appropriate for studying the macroscopic dynamics of large populations of spiking neurons consisting of excitatory and inhibitory units. To our knowledge, such a remarkable equivalence of the dynamical states between a mean-field neural mass model and its ground-truth spiking network model has not been demonstrated before. Our analysis shows that mean-field models are useful for quickly exploring the parameter space in order to predict states and parameters of the neural network they are derived from. Since the dynamical landscapes of both models are very similar, we believe that it should be possible to reproduce a variety of stationary and time-dependent properties of large-scale network simulations using low-dimensional population models. This may help to mechanistically describe the rich and plentiful observations in real neural systems when subject to stimulation with electric currents or

electric fields, such as switching between bistable *up* and *down-state* or phase locking and frequency entrainment of the population activity.

Bifurcations, as studied in dynamical systems theory, offer a plausible mechanism of how networks of neurons as well as the brain as a whole [58, 59] can change its mode of operation. Understanding the state space of real neural systems could be beneficial for developing electrical stimulation techniques and protocols, represented as trajectories in the dynamical landscape, which could be used to reach desirable states or specifically inhibit pathological dynamics.

Due to the variety of possible macroscopic network states that arise from this basic E-I architecture, it is critical to consider the state of the system in order to comprehend and predict its response to external stimuli. This might explain the numerous seemingly inconclusive experimental results from noninvasive brain stimulation studies [5, 6] where it is hard to account for the state of the brain before stimulation. In conclusion, additional to the stimulus parameters, the response of a system to external stimuli has to be understood in context of the dynamical state of the unperturbed system [3, 42, 60].

## Materials and methods

### Neural population setting

In order to derive the mean-field description of an AdEx network, we consider a very large number of $N \rightarrow \infty$ neurons for each of the two populations $E$ and $I$. We assume (1) random connectivity (within and between populations), (2) sparse connectivity [61, 62], but each neuron having a large number of inputs [63] $K$ with $1 \ll K \ll N$, (3) and that each neuron's input can be approximated by a Poisson spike train [64, 65] where each incoming spike causes a small ($c/J \ll 1$) and quasi-continuous change of the postsynaptic potential (PSP) [66] (*diffusion approximation*).

### The spiking neuron model

The adaptive exponential (AdEx) integrate-and-fire neuron model forms the basis for the derivation of the mean-field equations as well as the spiking network simulations. Each population $\alpha \in \{E, I\}$ has $N_\alpha$ neurons which are indexed with $i \in [1, N_\alpha]$. The membrane voltage of neuron $i$ in population $\alpha$ is governed by

$$C\frac{dV_i}{dt} = I_{\text{ion}}(V_i) + I_i(t) + I_{i,\text{ext}}(t), \tag{1}$$

$$I_{\text{ion}}(V) = g_L(E_L - V) + g_L \Delta_T \exp\left(\frac{V - V_T}{\Delta_T}\right) - I_A(t). \tag{2}$$

The first term of $I_{\text{ion}}$ (Eq 2) describes the voltage-depended leak current, the second term the nonlinear spike initiation mechanism, and the last term $I_A$, the somatic adaptation current. $I_{i,\text{ext}}(t) = \mu^{\text{ext}}(t) + \sigma^{\text{ext}} \xi_i(t)$ is a noisy external input. It consists of a mean current $\mu^{\text{ext}}(t)$ which is equal across all neurons of a population and independent Gaussian fluctuations $\xi_i(t)$ with standard deviation $\sigma^{\text{ext}}$ ($\sigma^{\text{ext}}$ is equal for all neurons of a population). For a neuron in population $\alpha$, synaptic activity induces a postsynaptic current $I_i$ which is a sum of excitatory and inhibitory contributions:

$$I_i(t) = C(J_{\alpha E} s_{i,\alpha E}(t) + J_{\alpha I} s_{i,\alpha I}(t)), \tag{3}$$

with $C$ being the membrane capacitance and $J_{\alpha\beta}$ the coupling strength from population $\beta$ to $\alpha$, representing the maximum current when all synapses are active. The synaptic dynamics is given by

$$\frac{ds_{i,\alpha\beta}}{dt} = -\frac{s_{i,\alpha\beta}}{\tau_s} + \frac{c_{\alpha\beta}}{J_{\alpha\beta}}(1 - s_{i,\alpha\beta})\sum_j G_{ij}\sum_k \delta(t - t_j^k - d_{\alpha\beta}). \tag{4}$$

$s_{i,\alpha\beta}(t)$ represents the fraction of active synapses from population $\beta$ to $\alpha$ and is bound between 0 and 1. $G_{ij}$ is a random binary connectivity matrix with a constant row sum $K_\alpha$ and connects neurons $j$ of population $\beta$ to neurons $i$ of population $\alpha$. With the constraint of a constant in-degree $K_\alpha$ of each unit, all neurons of population $\beta$ project to neurons of population $\alpha$ with a probability of $p_{\alpha\beta} = K_\alpha/N_\beta$ and $\alpha, \beta \in \{E, I\}$. $G_{ij}$ is generated independently for every simulation. The first term in Eq 4 is an exponential decay of the synaptic activity, whereas the second term integrates all incoming spikes as long as $s_{i,\alpha\beta} < 1$ (i.e. some synapses are still available). The first sum is the sum over all afferent neurons $j$, and the second sum is the sum over all incoming spikes $k$ from neuron $j$ emitted at time $t^k$ after a delay $d_{\alpha\beta}$. If $s_{i,\alpha\beta} = 0$, the amplitude of the postsynaptic current is exactly $C \cdot c_{\alpha\beta}$ which we set to physiological values from *in vitro* measurements [67] (see Table 1).

For neurons $i$ of the excitatory population, the adaptation current $I_{A,i}(t)$ is given by

$$\tau_A \frac{dI_{A,i}}{dt} = a(V_i - E_A) - I_{A,i}, \tag{5}$$

$a$ representing the subthreshold adaptation and $b$ the spike-dependent adaptation parameters. The inhibitory population doesn't have an adaptation mechanism, which is equivalent to setting these parameters to 0. When the membrane voltage crosses the spiking threshold, $V_i \geq V_s$, the voltage is reset, $V_i \leftarrow V_r$, clamped for a refractory time $T_{\text{ref}}$, and the spike-triggered adaptation increment is added to the adaptation current, $I_{A,i} \leftarrow I_{A,i} + b$. All parameters are given in Table 1.

Finally, we define the mean firing rate of neurons in population $\alpha$ as

$$r_\alpha(t) = \frac{1}{N_\alpha}\frac{1}{dt}\sum_{i=0}^{N_\alpha}\int_t^{t+dt}\delta(t' - t_i^k)dt, \tag{6}$$

which measures the number of spikes in a time window $dt$, set to the integration step size in our numerical simulations.

## The mean-field neural mass model

For a sparsely connected random network of AdEx neurons as defined by Eqs 1–5, the distribution of membrane potentials $p(V)$ and the mean population firing rate $r$ can be calculated using the Fokker-Planck equation in the thermodynamic limit $N \to \infty$ [13, 68]. Determining the distribution involves solving a partial differential equation, which is computationally demanding. A low-dimensional linear-nonlinear cascade model [24, 69] can be used to capture the steady-state and transient dynamics of a population in form of a set of simple ODEs. Briefly, for a given mean membrane current $\mu_\alpha$ with standard deviation $\sigma_\alpha$, the mean of the membrane potentials $\bar{V}_\alpha$ as well as the population firing rate $r_\alpha$ in the steady-state can be calculated from the Fokker-Plank equation [70] and captured by a set of simple nonlinear transfer functions $\Phi(\mu_\alpha, \sigma_\alpha)$ (shown in Fig 7a and 7b).

These transfer functions can be precomputed (once) for a specific set of single AdEx neuron parameters. All other parameters, such as input currents, network parameters and synaptic

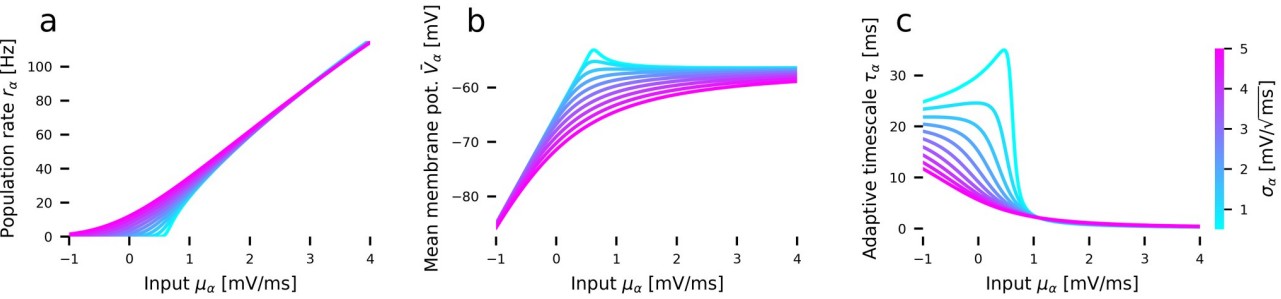

**Fig 7. Precomputed quantities of the linear-nonlinear cascade model. (a)** Nonlinear transfer function $\Phi$ for the mean population rate (Eq 15) **(b)** Transfer function for the mean membrane voltage (Eq 10) **(c)** Time constant $\tau_\alpha$ of the linear filter that approximates the linear rate response function of AdEx neurons (Eq 7). The color scale represents the level of the input current variance $\sigma_\alpha$ across the population. All neuronal parameters are given in Table 1.

coupling strengths, as well as the parameters that govern the somatic adaptation mechanism, the membrane timescale, and the synaptic timescale are identical and directly represented in the equations of the mean-field model. Thus, for any given parameter configuration of the AdEx network, there is a direct translation to the parameters that define the mean-field model. This also allows for direct comparison of both models under changes of said parameters.

The reproduction accuracy of the linear-nonlinear cascade model for a single population has been systematically reviewed in Ref. [23] and has been shown to reproduce the dynamics of an AdEx network in a range of different input regimes quite successfully, while offering significant increase in computational efficiency.

**Rate equations.** The derivation of the equations that govern the mean $\mu_\alpha$ and variance $\sigma_\alpha^2$ of the membrane currents, the mean adaptation current $\bar{I}_A$, and the mean $\bar{s}_{\alpha\beta}$ and variance $\sigma_{s,\alpha\beta}^2$ of the synaptic activity are presented further below. The full set of equations of the mean-field model reads:

$$\tau_\alpha \frac{d\mu_\alpha}{dt} = \mu_\alpha^{\text{syn}}(t) + \mu_\alpha^{\text{ext}}(t) - \mu_\alpha(t), \tag{7}$$

$$\mu_\alpha^{\text{syn}}(t) = J_{\alpha E}\bar{s}_{\alpha E}(t) + J_{\alpha I}\bar{s}_{\alpha I}(t), \tag{8}$$

$$\sigma_\alpha^2(t) = \sum_{\beta \in \{E,I\}} \frac{2J_{\alpha\beta}^2 \, \sigma_{s,\alpha\beta}^2(t) \, \tau_{s,\beta}\tau_m}{(1 + r_{\alpha\beta}(t)) \, \tau_m + \tau_{s,\beta}} + \sigma_{\text{ext},\alpha}^2, \tag{9}$$

$$\frac{d\bar{I}_A}{dt} = \tau_A^{-1}\left(a(\bar{V}_E(t) - E_A) - \bar{I}_A\right) + b \cdot r_E(t), \tag{10}$$

$$\frac{d\bar{s}_{\alpha\beta}}{dt} = -\tau_{s,\beta}^{-1} \, \bar{s}_{\alpha\beta}(t) + \left(1 - \bar{s}_{\alpha\beta}(t)\right) \cdot r_{\alpha\beta}(t), \tag{11}$$

$$\frac{d\sigma_{s,\alpha\beta}^2}{dt} = (1 - \bar{s}_{\alpha\beta}(t))^2 \cdot \rho_{\alpha\beta}(t) + \tau_{s,\beta}^{-2}\left(\rho_{\alpha\beta}(t) - 2\tau_{s,\beta}(\rho_{\alpha\beta}(t) + 1)\right) \cdot \sigma_{s,\alpha\beta}^2(t), \tag{12}$$

for $\alpha, \beta \in \{E, I\}$. All parameters are listed in Table 1. The mean $r_{\alpha\beta}$ and the variance $\rho_{\alpha\beta}$ of the

effective input rate from population $\beta$ to $\alpha$ for a spike transmission delay $d_{\alpha\beta}$ are given by

$$r_{\alpha\beta}(t) = \frac{c_{\alpha\beta}}{J_{\alpha\beta}} K_\beta \cdot r_\beta(t - d_\alpha), \tag{13}$$

$$\rho_{\alpha\beta}(t) = \frac{c_{\alpha\beta}}{J_{\alpha\beta}} \cdot r_{\alpha\beta}(t). \tag{14}$$

$r_\alpha$ is the instantaneous population spike rate, $c_{\alpha\beta}$ defines the amplitude of the post-synaptic current caused by a single spike (at rest, $s_{\alpha\beta} = 0$) and $J_{\alpha\beta}$ sets the maximum membrane current generated when all synapses are active (at $s_{\alpha\beta} = 1$).

To account for the transient dynamics of the population to a change of the membrane currents, $\mu_\alpha$ can be integrated by convolving the input with a linear response function. This function is well-approximated by a decaying exponential [23, 24, 69] with a time constant $\tau_\alpha$ (shown in Fig 7c). Thus, the convolution can simply be expressed as an ODE (Eq 7) with an input-dependent adaptive timescale $\tau_\alpha$ that is updated at every integration timestep. In Eq 7, $\mu_\alpha^{\mathrm{syn}}(t)$, as defined by Eq 8, represents the mean current caused by synaptic activity and $\mu_\alpha^{\mathrm{ext}}(t)$ the currents caused by external input.

The instantaneous population spike rate $r_\alpha$ is determined using the precomputed nonlinear transfer function

$$r_\alpha = \Phi(\mu_\alpha, \sigma_\alpha). \tag{15}$$

The transfer function $\Phi$ is shown in Fig 7a. It translates the mean $\mu_\alpha$ as well as the standard deviation $\sigma_\alpha$ (Eq 9) of the membrane currents to a population firing rate. Using an efficient numerical scheme [24, 70], this function was previously computed [23] from the steady-state firing rates of a population of AdEx neurons given a particular input mean and standard deviation. The transfer function depends on the parameters of the single AdEx neuron. Eq 10 governs the evolution of the mean adaptation current of the excitatory population. Eqs 11 and 12 describe the mean and the standard deviation of the fraction of active synapses caused by incoming spikes from population $\beta$ to population $\alpha$.

**Synaptic model.**  Following Ref. [71], we derive ODE expressions for the population mean $\bar{s}_{\alpha\beta}$ and variance $\sigma_{s,\alpha\beta}^2$ of the synaptic activity. We rewrite the synaptic activity given by Eq 4 of a neuron $i$ from population $\alpha$ caused by inputs from population $\beta$ with $\alpha, \beta \in \{E, I\}$ in terms of a continuous input rate $r_\beta$ (*diffusion approximation*) such that

$$\tau_{s,\beta} \frac{ds_{i,\alpha\beta}}{dt} = -s_i + \frac{c_{\alpha\beta}}{J_{\alpha\beta}}(1 - s_i)\left(K_\alpha r_\beta(t - d_\alpha) + \sqrt{K_\alpha r_\beta(t - d_\alpha)}\xi_i(t)\right), \tag{16}$$

with $K_\alpha = \Sigma_{j\in\alpha} G_{ij}$ being the constant in-degree of each neuron, $r_\beta(t - d_\alpha)$ the incoming delayed mean spike rate from all afferents of population $\beta$, and $\xi_i(t)$ being standardized Gaussian white noise. The current $I_i(t)$ of a neuron in population $\alpha$ due to synaptic activity is given by

$$I_i(t) = \sum_{\beta\in\{E,I\}} CJ_{\alpha\beta} s_{i,\alpha\beta}(t). \tag{17}$$

We split the mean from the variance of Eq 16 by first taking the mean over neurons of Eq 16. The mean synaptic activity $\bar{s}_{\alpha\beta} := \langle s_{i,\alpha\beta}\rangle_i$ of population $\alpha$ caused by input from population $\beta$ is then given by Eq 11. We get the differential equation of the variance $\sigma_{s,\alpha\beta}^2$ of $s_{i,\alpha\beta}$ in Eq 12 by applying Ito's product rule [72] on $d(s_{i,\alpha\beta}^2)$ and taking its time derivative.

**Input currents.**  Additional to the mean currents (Eq 7) in the population, we also keep track of their variance. Fig 7 shows the population firing rate and mean membrane potential

for different levels of variance of the membrane currents. Especially the adaptive time constant $\tau_\alpha$, which affects the temporal dynamics of the population's response, strongly depends on the variance of the input. Please note that the adaptive time constant $\tau_\alpha$ is not related to the somatic adaptation mechanism. Without loss of generality, we derive the variance of the membrane currents caused by a single afferent population $\alpha$ and later add up the contributions of two coupled populations (excitatory and inhibitory). Assuming that every neuron receives a large number of uncorrelated inputs (*white noise approximation*), we write the synaptic current $I_{i,\alpha}(t)$ in terms of contributions to the population mean and the variance [13, 73–76]:

$$I_{i,\alpha}(t) = C(\mu_\alpha(t) + \sigma_\alpha(t)\xi_i(t)). \tag{18}$$

In order to obtain the contribution of synaptic input to the mean and the variance of membrane currents, we (1) neglect the exponential term of $I_{\text{ion}}$ in Eq 2 and (2) assume that the membrane voltages are mostly subthreshold such that we can neglect the nonlinear reset condition. Numerical simulations have proven that these assumptions are justifiable in the parameter ranges that we are concerned with [71]. We apply these simplifications only in this step of the derivation. The exponential term, the neuronal parameters within it, and the reset condition still affect the precomputed functions (shown in Fig 7) and thus the overall population dynamics.

We substitute both approximations Eqs 17 and 18 separately into the membrane voltage Eq 1 and apply the expectation operator on both sides, which leads to two equations describing the evolution of the mean membrane potential. If we require that both approximations should yield the same mean potential $\langle V_{i,\alpha} \rangle$, we can easily see that $\mu_\alpha^{\text{syn}}(t) = J_{\alpha\alpha}\langle s_{i,\alpha\alpha}(t)\rangle$. Using Ito's product rule [72] on $dV^2$ and requiring that both approximations should also result in the same evolution of the second moment $\langle V_\alpha^2 \rangle$, we get

$$\sigma_\alpha^2(t) = 2J_{\alpha\alpha}(\langle V_{i,\alpha}s_{i,\alpha\alpha}\rangle - \langle V_{i,\alpha}\rangle\langle s_{i,\alpha\alpha}\rangle). \tag{19}$$

Taking the time derivative of Eq 19 and substituting the time derivative of $\langle V_\alpha s_{\alpha\alpha}\rangle$ by applying Ito's product rule on $d(V_\alpha s_{\alpha\alpha})$ we obtain

$$\frac{d\sigma_\alpha^2}{dt} = 2J_{\alpha\alpha}^2\sigma_{s,\alpha\alpha}^2(t) - \left(\frac{c\tau_\alpha K r_{\alpha\alpha}(t) + 1}{\tau_{s,\alpha}} + \frac{1}{\tau_m}\right)\sigma_{s,\alpha\alpha}^2(t) \tag{20}$$

Here, $\sigma_{s,\alpha\alpha}^2 := \langle s_{i,\alpha\alpha}^2\rangle - \langle s_{i,\alpha\alpha}\rangle^2$. The timescale of Eq 20 is much smaller than $\tau_\alpha$ of Eq 7. We can therefore approximate $\sigma_\alpha^2(t)$ well with its steady-state value:

$$\sigma_\alpha^2(t) = \frac{2J_{\alpha\alpha}^2\tau_m\tau_\alpha\sigma_{s,\alpha\alpha}^2(t)}{(c\tau_\alpha K r_{\alpha\alpha}(t) + 1)\tau_m + \tau_{s,\alpha}}, \tag{21}$$

$\tau_m = C/g_L$ being the membrane time constant. Adding up the variances in Eq 21 of both E and I subpopulations and the variance of the external input $\sigma_{\text{ext},\alpha}^2$, the total variance of the input currents is then given by Eq 9. The two moments of the membrane currents, $\mu_\alpha$ and $\sigma_\alpha$, fully determine the instantaneous firing rate $r_\alpha = \langle r_i\rangle_i$ (Eq 15), the mean membrane potential $\bar{V}_\alpha := \langle V_i\rangle_i$, and the adaptive timescale $\tau_\alpha$ (Fig 7).

**Adaptation mechanism.** The large difference of timescales of the slow adaptation mechanism mediated through K$^+$ channel dynamics compared to the faster membrane voltage dynamics [77, 78] and the synaptic dynamics allows for a separation of timescales [37] (*adiabatic approximation*). Therefore, each neuron's adaptation current can be approximated by its population average $\bar{I}_A$, which evolves according to Eq 10, where $a$ is the sub-threshold adaptation and $b$ is the spike-triggered adaptation parameter. $\bar{V}_\alpha(t) = \bar{V}_\alpha(\mu_\alpha, \sigma_\alpha)$ is the mean of the

membrane potentials of the population and was precomputed and is read from a table (Fig 7b) at every timestep. In the case of $a, b > 0$, i.e. when adaptation is active, we subtract the current $\bar{I}_A$ caused by the adaptation mechanism from the current $C \cdot \mu_\alpha$ caused by the synapses in order to obtain the net input current. The resulting firing rate of the excitatory population is then determined by evaluating $r_E = \Phi(\mu_E - \bar{I}_A/C, \sigma_E)$. For inhibitory neurons, adaptation was neglected ($a = b = 0$) since the adaptation mechanism was found to be much weaker than in the case of excitatory pyramidal cells [79].

## Obtaining bifurcation diagrams and determining bistability

Each point in the bifurcation diagrams in Figs 2 and 3 was simulated for pairs of external inputs $\mu_E^{\text{ext}}$ and $\mu_I^{\text{ext}}$ and the resulting time series of the excitatory population rate of the mean-field model and the AdEx network were analyzed and the dynamical state was classified.

To classify a point in the state space as *bistable* or not, in both models, we apply a negative and a subsequent positive stimulus to the excitatory population and measure the difference in activity after both stimuli are turned off again. In the AdEx network, a simple step input can cause over- and under-shoot as a reaction, which is a problem when assessing the stability of a basin of attraction around a fixed point. To overcome this problem, we constructed a slowly-decaying stimulus (in contrast to previous work [80], where bistability was identified using a step current). An inverted example of this stimulus is shown in Fig 4e and 4f. Using this stimulus, we first made sure that the population rate is in the *down-state* (the initial state) with an initial negative external input current that slowly decays back to zero. We then kicked the activity into the *up-state* (the target state) with a positive input and then let the current slowly decay to zero again. A slowly-decaying stimulus (in contrast to a step stimulus) ensures that transient effects such as over- and undershooting are minimized that would otherwise disturb the target state. As a result, the stability of the target state can be observed. We determined whether the *up-state* is stable or the activity has decayed back into the *down-state* by comparing the 1 s mean of the population rates after both stimuli have decayed. We classified a state as bistable if the rate difference after both kicks and subsequent relaxation phases was greater than 10 Hz. This threshold value was chosen to be smaller than every observed difference between the *up-* and the *down-state*. We confirmed the validity of this method for the mean-field model by using a continuation method to determine the stability of the fixed-point states, which provided the same bifurcation diagrams.

## Determining frequency spectra of the population activity

In the bifurcation diagrams in Figs 2 and 3, regions were classified as oscillating if the time series showed oscillations during the last 1 s after the first (negative) stimulus pushed the system into the *down-state*. The power spectrum of this oscillation was computed using the implementation of Welch's method [81] `scipy.signal.welch` in the Python package SciPy (1.2.1) [82]. A rolling Hanning window of length 0.5 s was used to compute the spectrum. If the dominant frequency was above 0.1 Hz and its power density was above 1 Hz we classified the state as oscillating. Visual observation of the time series confirmed that these thresholds classified the oscillating regions well. In cases where the transient of 1 s was too short such that the activity state of the population jumped from the *down-state* to the *up-state* within this period, misclassifications of these points as oscillatory states caused artifacts at the right-hand border of the *bistable* region to the *up-state*.

In Fig 5, we determined *frequency entrainment* by observing changes of the frequency spectrum of the population activity $r_E$. Each run was simulated for 6 s. We waited for 1 s for transient effects to vanish before turning on the oscillating stimulus and measured the power

spectrum of the remaining 5 s. A rolling Hanning window of length 1 s was used. For better visibility, the power was normalized between 0 and 1 on a logarithmic scale and plotted with a linear colormap.

### Measuring phase locking using the Kuramoto order parameter

In Fig 6 we quantified the degree of phase locking of an oscillatory input current with the E-I system's ongoing oscillation. We calculated the Kuramoto order parameter [83] to measure phase synchrony. The Kuramoto order parameter $R$ is given by:

$$R(t) = \frac{1}{N_{\mathrm{osc}}} \left| \sum_{j=1}^{N_{\mathrm{osc}}} e^{i\Phi_j(t)} \right|. \tag{22}$$

In our case, the number of oscillators $N_{\mathrm{osc}} = 2$ and $\Phi_j \in [0, 2\pi)$ is the instantaneous phase of the stimulus ($j = 1$) and the population activity $r_E$ ($j = 2$). We define the instantaneous phase $\Phi_j$ at time $t$ as

$$\Phi_j(t) = 2\pi \frac{t - t_n}{t_n - t_{n-1}}, \tag{23}$$

where $t_n$ is the time of the last maximum and $t_{n-1}$ the penultimate one. To robustly detect the oscillation maxima of the noisy AdEx network population rate, the time series was first smoothed using the Gaussian filter `scipy.ndimage.filters.gaussian_filter` implemented in SciPy. The Gaussian kernel had a standard deviation of 5 ms. Then, the maxima were detected using the peak finding algorithm `scipy.signal.find_peaks_cwt` with a peak-width between 0.1 and 0.2 ms.

For $R = 1$, perfect (zero-lag) phase synchronization is reached; if $R \approx 0$, the oscillations are maximally desynchronized. To measure *phase locking* in Fig 6a and 6c, we calculated the standard deviation in time of $R(t)$ after transient effects vanished for $t > 1.5$ s. A low standard deviation means that the phase difference between the input and the ongoing oscillation stays constant.

### Calculating equivalent electric field strengths

Our results can be used to estimate the necessary amplitude of an external electric field to reproduce the effects of electrical input currents. An external field at the location of a neural population might be produced by endogenous electric fields due to the activity of a neural population or external stimulation techniques such as transcranial electrical stimulation (tES) with direct (tDCS) or alternating (tACS) currents. The lack of a spatial extension of point neuron models such as the AdEx neuron makes it impossible to directly couple an external electric field that could affect the internal membrane voltages. Following Ref. [28], we obtain an equivalent electrical input *current* $C \cdot \mu_E^{\mathrm{ext}}(t)$ to a point neuron by matching it to reproduce the effects of an oscillating extracellular electric *field* on a spatially extended ball-and-stick (BS) model neuron of a given morphology. We calculated the equivalent current amplitudes for the exponential integrate-and-fire (EIF) neuron, which is the same as the AdEx neuron without somatic adaptation ($a = b = 0$). In the case with adaptation, the translation from current to field works for high frequency inputs only and the approximation breaks down for slowly oscillating inputs. Thus, we have limited our estimated field strengths to the case without adaptation.

The amplitude of the equivalent input *current* that causes the same subthreshold depolarization of a (linearized) EIF neuron as the somatic depolarization caused by the effects of an

oscillatory electric *field* on the BS neuron's dendrite is then calculated using

$$I_{\text{ext}} = A \left| \frac{U_{\text{BS}}(f)}{z_{\text{EIF}}(f)} \right|. \tag{24}$$

$A$ is the amplitude of the electric field in V/m, $U_{\text{BS}}(f)$ is the frequency-dependent polarization transfer function of the BS neuron and $z_{\text{EIF}}(f)$ is the impedance of the EIF neuron which are both given by

$$U_{\text{BS}}(f) = g_a(2e^{-z\,l_d} - \gamma)/\delta \tag{25}$$

$$z_{\text{EIF}}(f) = \frac{1}{g_L(1 - e^{\frac{V_r - V_T}{\Delta_T}}) + 2\pi i C \cdot f} \tag{26}$$

with the following substitutions:

$$w = 2\pi f, \qquad g_m = (\pi d_d)/\rho_m, \qquad g_a = (\pi(d_d/2)^2)/\rho_a \tag{27}$$

$$c_m = C_m d_d \pi, \qquad g_s = (\pi d_s^2)/\rho_s, \qquad c_s = C_m \pi d_s^2 \tag{28}$$

$$\alpha = \sqrt{\frac{g_m + \sqrt{g_m^2 + w^2 c_m^2}}{2g_a}}, \qquad \beta = \sqrt{\frac{-g_m + \sqrt{g_m^2 + w^2 c_m^2}}{2g_a}} \tag{29}$$

$$z = \alpha + i\beta, \qquad \gamma = 1 + exp(-2l_d z), \qquad \delta = \gamma(c_s wi + g_s) + zg_a(2 - \gamma). \tag{30}$$

The BS neuron we used to estimate electric field strengths has the following parameters: The soma has a diameter of $d_s = 10$ μm, a specific membrane capacitance of $C_m = 10$ mF/m$^2$ and a membrane resistivity of $\rho_s = 2.8$ Ωm$^2$. The dendritic cable has a length of $l_d = 1200$ μm, a diameter of $d_d = 2$ μm (as in the typical range of cortical pyramidal cells [32, 84]), a membrane resistivity of $\rho_m = 2.8$ Ωm$^2$ and an axial resistivity of $\rho_a = 1.5$ Ωm.

Using these parameters, an step input using an electric field with an amplitude of 1 V/m changes the somatic membrane potential by about 0.5 mV from its resting potential of −65 mV which is in agreement with *in vitro* measurements [32]. The curves shown in Fig 8 translate an electric field of a given amplitude and frequency to a corresponding input current and vice versa. An increase of the mean membrane current by 0.1 nA corresponds to an increase of the static electric field strength by 20 V/m.

## Numerical simulations

The mean-field equations were integrated using the forward Euler method. In Fig 2, each time series for a set of external inputs $\mu_E^{\text{ext}}$ and $\mu_I^{\text{ext}}$ in the bifurcation diagrams was obtained after t = 5 s simulation with an integration timestep of dt = 0.05 ms. In Fig 3 we simulated each point for t = 10 s with dt = 0.01 ms. For Fig 5 we simulated for t = 30 s with dt = 0.05 ms.

The spiking network model was implemented using BRIAN2 [85] (2.1.3.1) in Python. The equations were integrated using the implemented Heun's integration method. An integration step size of 1 ms was used. In all network simulations, $N_e = N_i$. For the bifurcation diagrams in Fig 2, we used N = $50 \times 10^3$ (i.e. $25 \times 10^3$ per population), a total simulation time of t = 6 s and in Fig 3, N = $20 \times 10^3$ and t = 6 s. The stimulation experiments in Fig 4 used N = $100 \times 10^3$, t = 3 s. The spectra in Fig 5 used N = $20 \times 10^3$, t = 6 s. Phase locking plots in Fig 6 used N = $20 \times 10^3$, t = 20 s.

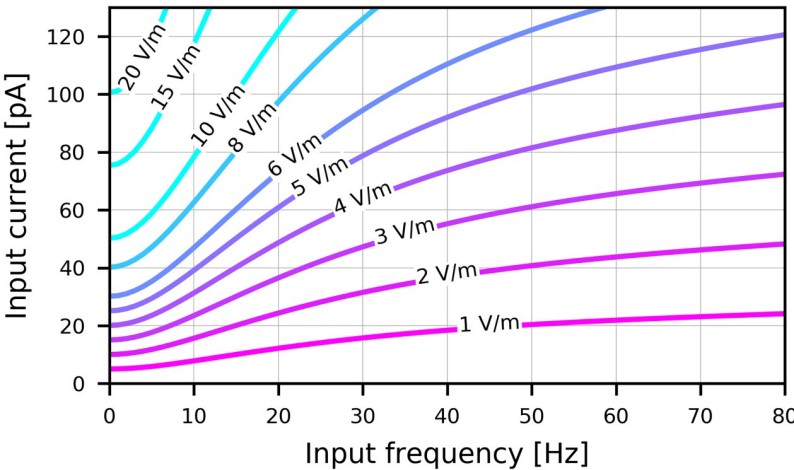

**Fig 8. Conversion between electric field amplitudes and equivalent input currents.** Each curve shows the frequency-dependent amplitude of an equivalent input current in pA to an exponential integrate-and-fire neuron with parameters as defined in Table 1 when the electric field amplitude acting on an equivalent ball-and-stick neuron is held constant. Electric field amplitudes in V/m are annotated for each curve.

Benchmarking the AdEx network with $N = 100 \times 10^3$ on a single core took around $10^4$ times longer to run than the corresponding mean-field simulation. This does not include the time required for initializing the simulation, such as setting up all synapses, which can also require a comparable amount of time. The computation time scales nearly linearly with the number of neurons.

An implementation of the mean-field model is available as a Python library in our GitHub repository https://github.com/neurolib-dev/neurolib. The Python code of the comparison to AdEx network simulations, the stimulation experiments, as well as the data analysis and the ability to reproduce all presented figures in this paper can be found at https://github.com/caglarcakan/stimulus_neural_populations.

## Supporting information

**S1 Fig. Spiking network activity and statistics. (a)** Population firing rate $r_E$ of the excitatory population in Hz (upper panels) and raster plots of 100 randomly chosen excitatory neurons (lower panels) in three different network states A2, A3 and B3, located in the bifurcation diagrams Fig 2. A2 is located in the fast excitatory-inhibitory limit cycle $LC_{EI}$, A3 in the high-activity asynchronous irregular *up-state*, and B3 in the adaptation-mediated slow limit cycle $LC_{aE}$. **(b)** The upper panel shows the distribution of coefficients of variation (CV) of the inter-spike-intervals (ISI) calculated as the variance of ISIs divided by the mean ISI of excitatory neurons for all three states. The lower panel shows spike count distributions. For each neuron, the spike count was calculated from the inverse of the mean of the ISI distribution. Simulations were run with $N = 100 \times 10^3$ neurons for 10s each. The statistics were computed for $t > 500$ ms for the neurons shown in (a). All parameters are given in Table 1.
(TIF)

**S2 Fig. Bifurcation diagrams with maximum rate of the inhibitory population. (a)** Bifurcation diagram of the mean-field model without adaptation with *up* and *down-states*, a bistable region *bi* (green dashed contour) and an oscillatory region $LC_{EI}$ (white solid contour). **(b)** Diagram of the corresponding AdEx network. **(c)** The mean-field model with somatic adaptation

has a slow oscillatory region $LC_{aE}$. **(d)** Diagram of the corresponding AdEx network. The color indicates the maximum population rate of the inhibitory population (clipped at 80 Hz). All parameters are given in Table 1.
(TIF)

**S3 Fig. Bifurcation diagrams depicting the difference of excitatory and inhibitory amplitudes. (a)** Bifurcation diagram of the mean-field model without adaptation with *up* and *down-states*, a bistable region *bi* (green dashed contour) and an oscillatory region $LC_{EI}$ (white solid contour). **(b)** Diagram of the corresponding AdEx network. **(c)** The mean-field model with somatic adaptation has a slow oscillatory region $LC_{aE}$. **(d)** Diagram of the corresponding AdEx network. The color indicates the difference of excitatory and inhibitory amplitudes (clipped from -100 Hz to 100 Hz). All parameters are given in Table 1.
(TIF)

**S4 Fig. Bifurcation diagrams with the dominant oscillation frequency of the excitatory population. (a)** Bifurcation diagram of the mean-field model without adaptation. **(b)** Diagram of the corresponding AdEx network. **(c)** Mean-field model with somatic adaptation. **(d)** Diagram of the corresponding AdEx network. The color indicates the difference of excitatory and inhibitory amplitudes (clipped at 35 Hz). All parameters are given in Table 1.
(TIF)

**S5 Fig. Bifurcation diagrams of the mean-field model for changing coupling strengths.**
Stacked bifurcation diagrams depending on the mean input current to populations E and I showing dynamical states for intervals of $J_{EE}$ and $J_{II}$ (outer axis), $J_{IE}$ and $J_{EI}$ (inner axis) by values of 0.5 V/ms. The middle rows and columns correspond to the default value of the corresponding parameter (see Table 1). White contours are oscillatory areas $LC_{EI}$, green dashed contours are bistable regions. Diagram in the middle (blue box) corresponds to bifurcation diagram Fig 2a. $a = b = 0$. For all other parameters, see Table 1.
(TIF)

**S6 Fig. Bifurcation diagrams of the AdEx network model for changing coupling strengths.**
Stacked bifurcation diagrams for a subset of the values depicted in S4 Fig depending on the mean input current to populations E and I showing dynamical states for changing $J_{EE}$ and $J_{II}$ (outer axis), $J_{IE}$ and $J_{EI}$ (inner axis) by intervals of 0.5 mV/ms. The middle rows and columns correspond to the default value of the corresponding parameter (see Table 1). In this figure, all of the four coupling parameters have been varied independently. Empty plots were not computed. White contours within the plots denote the boundaries of the oscillatory areas $LC_{EI}$, green dashed contours the boundaries of bistable regions. Position in the middle (blue box) corresponds to bifurcation diagram Fig 2b. Number of neurons $N = 20 \times 10^3$, $a = b = 0$. For all other parameters, see Table 1.
(TIF)

**S7 Fig. Finite-size effects on bifurcation diagrams of the AdEx network with increasing number of neurons N.** Bifurcation diagrams depict the state space of the E-I system without adaptation in terms of the mean external input currents $C \cdot \mu_\alpha^{ext}$ to both subpopulations $\alpha \in \{E, I\}$. *Up* (bright area) and *down-states* (dark blue area), a bistable region *bi* (green dashed contour) and an oscillatory region $LC_{EI}$ (white solid contour) are visible. All parameters are given in Tables 1 and 2.
(TIF)

**S8 Fig. Finite-size effects in the AdEx network on E-I oscillation amplitudes.** Oscillation amplitudes in the limit cycle $LC_{EI}$ fluctuate due to finite-size effects in the AdEx network.

The system is parameterized in point A1 and pushed into the limit cycle by a constant input as in Fig 4b. **(a)** Traces of the population firing rates are shown (black) with the oscillation's maxima marked (red dots) for an increasing number of neurons $N$ in each panel (excitatory plus inhibitory). **(b)** The left panel shows the mean amplitude and the standard deviation as a function of the population size $N$ on a semi-logarithmic scale. With increasing $N$, the amplitude of the oscillation decreases. The right panel shows the coefficient of variation (CV) of the amplitudes on a semi-logarithmic scale. The CV decreases with increasing number of neurons. Each point was measured from 20 realizations of 2 seconds of oscillatory activity. One randomly chosen realization for each $N$ is shown in (a). All parameters are given in Tables 1 and 2.
(TIF)

## Acknowledgments

We would like to thank Dr. Josef Ladenbauer and Dr. Moritz Augustin for their work on reduced population models, their mathematical guidance, and many insightful discussions. We would like to thank Dr. Florian Aspart his work on the effects of extracellular electric fields, for helping to incorporate the results in this article, and for a helpful exchange of ideas.

## Author Contributions

**Conceptualization:** Caglar Cakan, Klaus Obermayer.

**Formal analysis:** Caglar Cakan.

**Funding acquisition:** Klaus Obermayer.

**Methodology:** Caglar Cakan.

**Project administration:** Klaus Obermayer.

**Software:** Caglar Cakan.

**Supervision:** Klaus Obermayer.

**Validation:** Caglar Cakan.

**Visualization:** Caglar Cakan.

**Writing – original draft:** Caglar Cakan.

**Writing – review & editing:** Caglar Cakan, Klaus Obermayer.

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
