## [Decision Letter · Decision Letter 0]

20 Nov 2019

Dear Dr Cakan,

Thank you very much for submitting your manuscript 'Biophysically grounded mean-field models of neural populations under electrical stimulation' for review by PLOS Computational Biology. Your manuscript has been fully evaluated by the PLOS Computational Biology editorial team and in this case also by independent peer reviewers. The reviewers appreciated the attention to an important problem, but raised some substantial concerns about the manuscript as it currently stands. While your manuscript cannot be accepted in its present form, we are willing to consider a revised version in which the issues raised by the reviewers have been adequately addressed. We cannot, of course, promise publication at that time.

The reviewers make a number of helpful suggestions on how to make your work more comprehensible and solidify the claims you make. In particular, your paper would be improved by a more rigorous assessment of when the match between mean field theory and simulation break down, and whether the theory then holds for biologically relevant parameter settings. Additionally, given the focus on electrical stimulation, the discussion should take the opportunity to extract biological insight from the theoretical approach, for example by comparison with experimental findings or concrete experimental predictions (or ideally both).

Sincerely,

Abigail Morrison

Associate Editor

PLOS Computational Biology

Lyle Graham

Deputy Editor

PLOS Computational Biology

[LINK]

Reviewer's Responses to Questions

**Comments to the Authors:**

Reviewer #1: The paper of Cakan and Obermayer describes an interesting study of cortical network dynamics where a network of AdEx neurons is studied on two levels: (i) explicit representation of each AdEx neuron and (ii) a corresponding mean-field model. First, the phase-diagrams of the two models receiving DC-currents are investigated, demonstrating a close correspondence. Next, the response of these networks to oscillatory inputs at various frequencies are investigated. I find the work to be highly accomplished and interesting and recommend publication in PLoS Computational Biology. However, I have some points that the authors should address prior to publication.

1. The paper is “sold” as a study of electric field effects, while what is shown (in Figure 4-6) is results to oscillatory input currents. Building on previous work on the group (Aspart et al, 2016) it is argued that the effect of oscillatory electric fields can be modelled using such currents. This link assumes that the main effect of imposed electrical fields comes from the effect on dendrites, rather than axons. As as far as I know this is not established (see, for example, the work of Rattay which focuses on the effects of electrical fields on the axon stub attached to the soma). This limitation should be discussed and maybe it should be considered to change the title to mention “.. under electrical stimulation” to something like “receiving oscillatory input”.

2. I am also a bit uncertain about what is argued when it comes to electrical fields, are we only talking about externally imposed electrical fields or also ephaptic effects.

3. As I understand it, the mean field (MF) model assumes very weak synaptic couplings (c<<1), and in this regime it is observed good correspondence between MF og AdEx. However, in vivo synapses have been shown to have long tail distributions and thus strong synaptic couplings are abundant. This is a possible shortcoming of the model and the paper as it aims to be experimentally relevant. Simulations of the Mean Field model with varying coupling strengths are given in Fig S4, and it is mentioned in the discussion that bifurcation diagrams seems fairly robust. 1) A comparison with the AdEx model would strengthen these claims and possibly shed light on differences. 2) It would be interesting to see when/if similarities between AdEx and mean field brake when e.g. increasing synaptic couplings or making them heterogeneous

4. More insight into the differences that are apparent in many of the figures would serve the paper well.

For example:

* In Fig 4. a-b there seems to be a beating in the AdEx firing rate not apparent in mean field model

* In Fig 4. g-h there seems to be a re-bound after initial transient in the AdEx firing rate not apparent in mean field - this would also be interesting to visualize in frequency spectrum.

* Fig 6 shows AdEx vs Mean field phase locking, but also with calculated external fields. However, it is mentioned earlier that fields may only be calculated without adaptation, are these results thus without adaptation i.e. the exponential integrate and fire model, not the AdEx?

Other minor points:

* I couldn’t find the point B2 (mentioned in the text) in Figure 2.

* What is meant by “the biophysical parameters of the AdEx network model are preserved in the mean-field description”. I assume that there are not explicit formulas giving all mean-field parameters in terms of AdEx parameters?

Reviewer #2: The authors present a very interesting and systematic analysis of the mean-field model of populations of AdEx neurons derived in [1]. Other work [2] has shown that the parameter describing electrical stimuli in the mean-field model can be mapped to values used in experiments. The work in this manuscript thus builds on the previous work in [1] and [2] but lacks a strong conclusion. This work could benefit from either a sound embedding in experimental literature or a more clear cut theoretical question.

The authors show that the mean-field model captures the dynamical state of the high-dimensional system as well as state transitions induced by electrical stimulation. Specifically, the mean-field model:

- accounts for the bifurcations encountered when varying input to excitatory and inhibitory neurons.

- captures responses to transient and sustained input currents over a wide frequency range.

- captures phase locking between the population rate and the stimulus.

The authors systematically explore the dynamical states of populations of AdEx neurons and the mean-field model by varying the external input to excitation and inhibition. This work nicely shows that the mean-field model accounts for the dynamics of the high-dimensional model over a wide parameter range. However, the authors of [1] already showed that the mean-field model captures the oscillatory dynamics displayed by coupled populations of AdEx neurons.

This analysis also shows that stronger adaptation shrinks the region of bi-stability and replaces it with slow oscillations. It would be interesting to strengthen this nice mechanistic insight by theoretical analysis of the mean-field model. How does this result relate to other theoretical works of adaption? For example: [3] show that the single neuron adaptation time scale is not necessarily transferred to population dynamics and [4] show that the filter properties of the population dynamics induced by adaption are shaped by the microscopic state of the network, i.e. whether the network is in the chaotic regime.

The second main aspect of the manuscript is the direct link of the electrical external input in the mean-field description to electrical field strengths used in experiments. The conversion of the spatially extended electrical field to a point measure that can be used in the mean-field model is based on the derivations in [2]. The authors state that realistic electrical currents can initiate transitions between dynamical states in the mean-field model. However, as far as I can tell, there is neither a discussion nor a reference to experiments and it is therefore not clear what these realistic current values are based on. Similarly, the statement that weak electrical inputs to the brain affect brain dynamics is not explored or substantiated further in the manuscript.

My suggestions for improving the paper are:

- The work could be motivated in more detail.

- What is the motivation for analyzing frequency entrainment?

- Why do the authors consider phase locking between input and population rate? Why at this particular stimulus frequency range?

- The authors could explore differences between the microscopic dynamics of the AdEx neurons and the mean-field model.

- How heterogeneous are the microscopic dynamics?

- Are there finite-size effects?

Minor comments:

- what is the number of neurons used in the simulation?

- Author summary: double 'field'

- The frequency entrainment for the mean-field and the AdEx network - - look quite different (Fig. 5c,d). It would be interesting to comment on that.

- The time traces of Fig. 6c,d are zoomed out too far to see the phase locking.

- Line 425: double 'the'

- Line 539: out-> our

- Line 588: sentence does not finish

[1] Augustin M, Ladenbauer J, Baumann F, Obermayer K. Low-dimensional spike rate models derived from networks of adaptive integrate-and-fire neurons: comparison and implementation. PLOS Comput Biol, accepted. 2017;.  

[2] Aspart F, Ladenbauer J, Obermayer K. Extending Integrate-and-Fire Model Neurons to Account for the Effects of Weak Electric Fields and Input Filtering Mediated by the Dendrite. PLoS Computational Biology. 2016;12(11):1–29. doi:10.1371/journal.pcbi.1005206.

[3] Beiran M, Ostojic S (2019) Contrasting the effects of adaptation and synaptic filtering on the timescales of dynamics in recurrent networks. PLoS Comput Biol 15(3): e1006893. https://doi.org/10.1371/journal.pcbi.1006893

[4] Muscinelli SP, Gerstner W, Schwalger T (2019) How single neuron properties shape chaotic dynamics and signal transmission in random neural networks. PLoS Comput Biol 15(6): e1007122. https://doi.org/10.1371/journal.pcbi.1007122

**Have all data underlying the figures and results presented in the manuscript been provided?**

Reviewer #1: Yes

Reviewer #2: None

PLOS authors have the option to publish the peer review history of their article (what does this mean?). If published, this will include your full peer review and any attached files.

Reviewer #1: No

Reviewer #2: No

---

## [Decision Letter · Decision Letter 1]

28 Feb 2020

Dear Mr. Cakan,

Thank you very much for submitting your manuscript "Biophysically grounded mean-field models of neural populations under electrical stimulation" for consideration at PLOS Computational Biology. As with all papers reviewed by the journal, your manuscript was reviewed by members of the editorial board and by several independent reviewers. The reviewers appreciated the attention to an important topic. Based on the reviews, we are likely to accept this manuscript for publication, providing that you modify the manuscript according to the review recommendations.

Sincerely,

Abigail Morrison

Associate Editor

PLOS Computational Biology

Lyle Graham

Deputy Editor

PLOS Computational Biology

[LINK]

Reviewer's Responses to Questions

**Comments to the Authors:**

Reviewer #1: The authors have addressed all my comments and questions satisfactorily, and I think the paper is ready for publication.

Reviewer #2: The authors significantly improved the Introduction and Discussion section by highlighting their main achievements and embedding their work into the literature. They show the microscopic dynamics underlying population dynamics. They also considered finite-size effects and nicely showed that larger network size yields better agreement with the mean-field theory. Hence the authors addressed all my issues and I have just a couple of minor issues.

Minor:

line 441: in order comprehend -> to comprehend

Figure R6: please add the mean-field prediction for the amplitude

**Have all data underlying the figures and results presented in the manuscript been provided?**

Reviewer #1: None

Reviewer #2: Yes

PLOS authors have the option to publish the peer review history of their article (what does this mean?). If published, this will include your full peer review and any attached files.

Reviewer #1: Yes: Gaute Einevoll

Reviewer #2: No
---

## [Editor Report · Decision Letter 2]

24 Mar 2020

Dear Mr. Cakan,

We are pleased to inform you that your manuscript 'Biophysically grounded mean-field models of neural populations under electrical stimulation' has been provisionally accepted for publication in PLOS Computational Biology.

Best regards,

Abigail Morrison

Associate Editor

PLOS Computational Biology

Lyle Graham

Deputy Editor

PLOS Computational Biology

---

## [Editor Report · Acceptance letter]

8 Apr 2020

PCOMPBIOL-D-19-01462R2 

Biophysically grounded mean-field models of neural populations under electrical stimulation

Dear Dr Cakan,

I am pleased to inform you that your manuscript has been formally accepted for publication in PLOS Computational Biology. Your manuscript is now with our production department and you will be notified of the publication date in due course.

With kind regards,

Laura Mallard
